# Aqueous synthesis of highly functional, hydrophobic, and chemically recyclable cellulose nanomaterials through oxime ligation

**Elena Subbotina** [1] ✉, **Farsa Ram**[1], **Sergey V. Dvinskikh**[2], **Lars A. Berglund** [1] & **Peter Olsén** [1] ✉

Cellulose nanofibril (CNF) materials are candidates for the sustainable development of high mechanical performance nanomaterials. Due to inherent hydrophilicity and limited functionality range, most applications require chemical modification of CNF. However, targeted transformations directly on CNF are cumbersome due to the propensity of CNF to aggregate in non-aqueous solvents at high concentrations, complicating the choice of suitable reagents and requiring tedious separations of the final product. This work addresses this challenge by developing a general, entirely water-based, and experimentally simple methodology for functionalizing CNF, providing aliphatic, allylic, propargylic, azobenzylic, and substituted benzylic functional groups. The first step is $NaIO_4$ oxidation to dialdehyde-CNF in the wet cake state, followed by oxime ligation with O-substituted hydroxylamines. The increased hydrolytic stability of oximes removes the need for reductive stabilization as often required for the analogous imines where aldehyde groups react with amines in water. Overall, the process provides a tailored degree of nanofibril functionalization (2–4.5 mmol/g) with the possible reversible detachment of the functionality under mildly acidic conditions, resulting in the reformation of dialdehyde CNF. The modified CNF materials were assessed for potential applications in green electronics and triboelectric nanogenerators.

The worldwide production of petroleum-based plastics went from 50 million tons per year in 1976 to 368 million tons in 2019 (according to Statista). Increased greenhouse gas emissions and dwindling resources require a switch to a carbon-neutral materials society[1]. Cellulosic biomass has great potential for increased use in the large-scale production of materials replacing plastics[2]. Next-generation bio-based materials must have properties matching or transcending petroleum-based plastics in order to facilitate substitution. Cellulose nanofibrils (CNF) are interesting candidates and are obtained by defibrillation of cellulosic biomass by chemical or mechanical means, for use in materials with physical and mechanical properties superior to relevant plastics[3–5].

CNF from plant cell walls such as wood fibers is usually around 3–10 nm in diameter and 0.5–2 μm in length. Each nanofiber is semi-crystalline and consists of parallel cellulose molecules in extended chain conformation. For this reason, the modulus (100–130 GPa) and tensile strength (2–6 GPa) of cellulose nanofibers are very high. The

[1]Department of Fibre and Polymer Technology, Wallenberg Wood Science Center, KTH Royal Institute of Technology, Teknikringen 56, 100 44 Stockholm, Sweden. [2]Department of Chemistry, KTH Royal Institute of Technology, Teknikringen 30, 100 44 Stockholm, Sweden. ✉e-mail: elenasu@kth.se; polsen@kth.se

elementary cellulose fibril surface, covered with hydroxyl functionality, is explored for chemical modification toward targeted applications. CNF is only stable as dilute suspensions in water, and prone to coagulate in organic solvents or at higher concentrations. Due to an aqueous environment (often incompatible with common organic reactions), the heterogeneous nature of the substrate, and high dilution any chemical transformations and purifications of CNF are experimentally challenging.

Apart from methods relying on the dissolution of cellulose where the nanostructure is completely lost, (dimethylsulfoxide/tetrabutylammonium fluoride (DMSO/TBAF)[6], lithium chloride/N,N-dimethylacetamide solution (LiCl/DMAc)[7], DMSO/1,8-Diazabicyclo(5.4.0)undec-7-ene (DBU)/CO$_2$[8], and ionic liquids[9–11]) majority of methods are applied to cellulose fibers followed by their defibrillation to individual nanofibers. These modifications are inherently inhomogeneous due to the poor availability of cellulose chains in the central part of the fibers. These methods often aim to introduce charges on cellulose chains to ease the defibrillation and can proceed through oxidation, such as 2,2,6,6-tetramethyl-1-piperidinyloxy (TEMPO) oxidation[4,12], or chemical modification via carboxymethylation[13–15], sulfonation[16,17], phosphorylation[18], and more recently succinylation[19]. Surface hydrophobization is another commonly implemented method often achieved through esterification with acid anhydrides[20–23], acyl chlorides[24] or other acyl derivatives[25] and silylation using chlorosilanes or alkoxysilanes[26–28] in organic solvents, generally requiring large excess of reagents[20,21]. For these transformations, the solvent exchange is generally required, even though there are acetylation methods that tolerate residual water[25]. Solvent exchange, apart from the obvious environmental and practical burden, limits the degree of functionalization through decreased functional group accessibility and may also dissolve highly modified carbohydrate segments on the CNF[26].

The optimal reaction solvent for CNF modification would be water. Water increases the chemical accessibility of cellulose chains without destroying the nanostructure, and is also beneficial from a green chemistry point of view. Exampefied strategies for CNF modification with water as a solvent include; amide coupling of cellulose carboxyl groups using 1-ethyl-3-[3-dimethylaminopropyl] carbodiimide (EDC) and N-Hydroxysuccinimide (NHS) to graft, e.g., amines[29], amino-terminated PEG[30], cysteine[31] on TEMPO-CNF and carboxymethyl cellulose[32]. However, these methods require stoichiometrically matched amounts of reagents (EDC, NHS) and generate a stoichiometric amount of waste, which complicates the purification process and compromises the atom efficiency of the transformation. Other examples include Cu(I)-catalyzed azide-alkyne click-reactions where azide functionality is installed either via epoxidation[33] or silylation[34]. Silylation, while being a promising example of a water-based modification pathway, can be complicated due to the propensity of trialkyl silanes to hydrolysis and oligomerization which can result in less defined functionalization, where oligomeric siloxanes can be either covalently attached or absorbed on the surface of cellulosic material[35,36]. Purification of the final material is another concern. In the abovementioned methodologies, purification of modified CNF is performed either via dialysis, which is very time-consuming or via centrifugation followed by redispersion of nanofibers, which can cause aggregation of individual fibrils. In addition, the modifications are irreversible at standard conditions. Thus, harsh reaction conditions are needed to detach functional groups from surface-modified CNF (e.g., hydrolysis of amides) or a stoichiometric reagent (e.g., fluoride source for silyl ethers). The possibility of mild "on-demand" detachment of the covalently linked functional groups would enable the recycling of the chemicals and significantly improve the sustainability and economic feasibility of modified CNF in targeted applications.

In this work, we aim to develop a modification strategy applicable directly to cellulose nanofibers, which is operational in water, retains the nanostructure, and is reversible, general, and experimentally simple. We turned our attention to periodate (NaIO$_4$) oxidation of cellulose, which is a powerful method to introduce aldehyde moieties on cellulose anhydroglucose units (AGUs) via oxidative cleavage of C2−C3 bond[37–41]. While being a well-known method for the functionalization of cellulose fibers, examples of periodate oxidations performed on nanofibers are rare and are performed on suspensions of either carboxymethylated CNF[42] or TEMPO-CNF[39], where the modified products were used as paper strengthening additives, rather than individual materials. When it comes to using aldehyde handles for further modifications, the majority of protocols are based on reductive amination, where the aldehyde is captured by a significant excess of amine followed by (or performed in situ) reduction, most commonly with NaBH$_4$[43]. This strategy has been used to attach fatty amines[44,45], ethylenediamine[46], or multifunctional amines[47] to cellulose nanocrystals (CNC). A reduction step is necessary due to the hydrolytic instability of imines in water. In addition to the apparent disadvantages of an extra step, in situ reductions of imines are generally complicated by the competing reduction of aldehydes, resulting in a low degree of functionalization (e.g., 11–24% of aldehyde groups were modified[44]). The reaction is exothermic, with hydrogen gas as a possible byproduct, even though optimization of the reaction conditions can minimize this side reaction. And importantly, the generated amine group is highly stable, limiting the recovery and recycling of the functionalized CNF.

To overcome the above-listed challenges, we envisioned an approach where NaIO$_4$ oxidation of Holo-CNF (CNF with relatively high content of hemicelluloses) with low surface charge is carried out on a preformed wet cake. The wet cake is obtained via a vacuum filtration of CNF suspension leading to a random in-plane assembly of nanofibers in the final material. This prevents nanofibers from agglomeration, allows for a high concentration of reagents which makes the reaction significantly faster, eliminates any tedious separations, and allows for the recovery and regeneration of the oxidant. A low-charge substrate prevents anionic repulsion between the cellulose chain and IO$_4^-$ anions. For further modification of the formed aldehyde groups, we chose salts of O-substituted hydroxylamines, resulting in oxime linkages, which are more stable towards hydrolysis compared to imines and thus might not require reductive stabilization (Fig. 1)[48,49]. Although formed oximes are stable in water at a wide pH range, the reaction is still reversible under acidic conditions in the presence of a trapping agent (ketone or aldehyde to trap the released hydroxylamine, Fig. 1). We hypothesized that oximes would have suitable chemical characteristics for a modular and reversible derivatization platform of CNF, with water as the only reaction medium.

## Results
### Facile and controlled oxidation of CNF to DA-CNF
The initial focus was to develop an efficient and operationally simple method for the oxidation of CNF, facilitating the homogeneous distribution of aldehyde groups on the nanofiber. Most procedures to oxidize cellulose with NaIO$_4$ are performed in suspensions of millimeter-long cellulose fibers followed by mechanical defibrillation[41,50]. Covalent hemiacetal and acetal linkages formed upon the oxidation between aldehyde and hydroxyl groups of cellulose will complicate defibrillation[51]. Moreover, the oxidation of large cellulosic wood fibers may lead to the heterogeneous distribution of the oxidized fragments. NaIO$_4$ oxidation of suspensions of nanofibers could overcome these issues but requires a low concentration of nanofibers to avoid aggregation. This leads to long reaction times and tedious separation of the oxidant from the reaction mixture (often via dialysis, taking several days)[42,52].

To overcome the listed limitations, we designed a protocol allowing for the oxidation of cellulose nanofibrils at high concentrations yet avoiding aggregation and heterogeneous oxidation. We carried out the reaction on a pre-made CNF wet-cake, which forms by vacuum filtration of a dilute suspension of CNF in water. This allows for uniform (random in-plane) distribution of the nanofibrils[3]. We selected

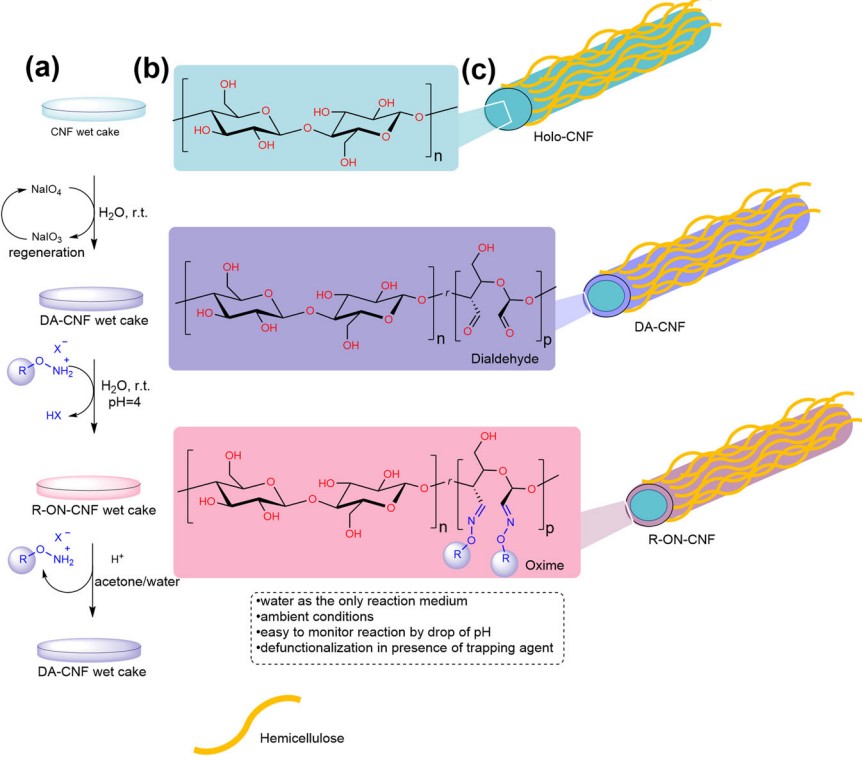

**Fig. 1 | Schematic representation of the developed method. a** Oxidation and functionalization of Holo-CNF[5] in water. **b** Schematic representation of molecular structure of pristine Holo-CNF, DA-CNF, and R-ON-CNF with regions of pristine cellulose and modified cellulose chains. **c** Schematic representation of nanostructure of Holo-CNF, DA-CNF, and R-ON-CNF with predominant modification at the surface of the nanofibrils. Yellow "polymers" represent glucomannan and xylan hemicelluloses[5].

Holo-CNF prepared via mild peracetic acid delignification of wood, which results in preserved hemicelluloses (glucomannan and xylan) in the final material[5]. Oxidation is performed by immersion of a Holo-CNF wet-cake in a solution of NaIO₄ giving rise to dialdehyde-CNF (DA-CNF). The DA-CNF wet-cake is removed from the oxidant solution, washed with water, and is directly applicable for further transformations. Using this approach, we alleviate a separation of CNF from the oxidant and enable the spent solution of the oxidant to be reused in multiple consecutive oxidation steps (although the active form of the oxidant−NaIO₄ shows decreased concentration and the reduced form −NaIO₃ shows increased concentration so that reaction times will be longer). There are efficient and benign methods to generate NaIO₄ from NaIO₃ with, e.g., ozone[53] or electrochemically[54,55], providing potential for a closed-loop oxidative system. The desired degree of oxidation is achieved by variations in reaction time or concentration of the oxidant (Fig. 2, Supplementary Table 1). As the oxidation of the Holo-CNF wet cake progress, its diameter uniformly decreases, while the thickness increases (Supplementary Table 1). Even at a high degree of oxidation, none of the aldehyde groups were observed by solid-state carbon NMR. The absence of shifts above 200 ppm suggests that the aldehydes are hydrated or participate in hemiacetal formation with residual hydroxyls in the carbohydrates. As such, shrinkage is believed to be due to a combination of hemiacetal formation and transition from strong in-plane CNF orientation to a state where CNF fibrils contract in-plane and possibly become wavy with out-of-plane oriented segments (thickness increase). This is probably caused by decreased cellulose crystallinity, resulting in less rod-like CNF. We found that after 7.5 h crystallinity of the samples was completely lost, suggesting, that the oxidation occurred not only on the surface of CNF but within inner regions as well. The proposed CNF organization is supported by SEM images of fracture surfaces of freeze-dried DA-CNF wet cakes (Fig. 2). It is important to mention, that part of aldehyde

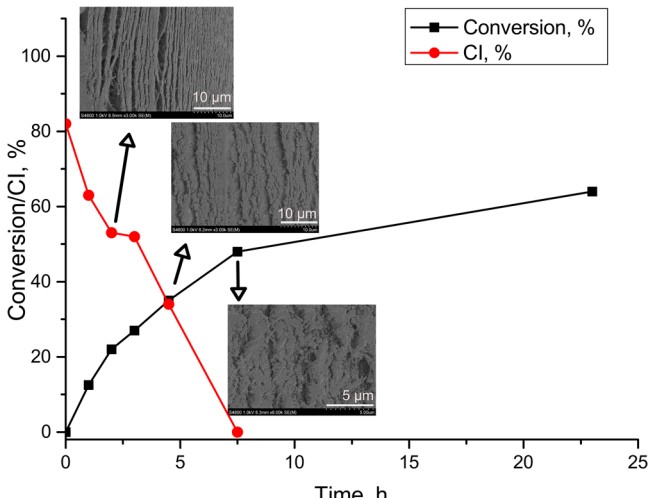

**Fig. 2 | Oxidation of Holo-CNF wet cake using NaIO₄ solution (2.2 g of NaIO₄ in 40 mL H₂O, ca. 250 mg of Holo-CNF (dry weight)).** Correlation between the reaction time, conversion (black squares), and crystallinity index (CI, red circles, Supplementary Fig. 37). SEM images of fracture surfaces of freeze-dried DA-CNF wet cakes with different degrees of oxidation. Source data are provided as a Source Data file.

groups can be located on hemicellulose chains, which cover cellulose nanofibrils in Holo-CNF. Using TEMPO-CNF in this transformation is impractical due to the low reaction rate. The reason is the repulsion of the IO₄⁻ anion by the negatively charged carboxyls at TEMPO-CNF surfaces. In summary, it is believed that the hemicellulose-covered surface of Holo-CNF both facilitates chemical accessibility and rate of

**Table 1 | Static mechanical properties of CNF and DA-CNF films with different degrees of oxidation obtained upon different reaction times**

| Entry | Sample (aldehyde mmol/g) | Time (h) | Tensile strength (MPa) | Elongation at break (%) | Young's Modulus (GPa) | Thickness (mm) |
|---|---|---|---|---|---|---|
| 1 | Holo-CNF | 0 | 270 ± 22 | 5.2 ± 0.5 | 12 ± 1 | 0.033 |
| 2 | DA-CNF (2.6) | 2 | 197 ± 11 | 4 ± 1 | 14 ± 1 | 0.038 |
| 3 | DA-CNF (3.2) | 3 | 180 ± 9 | 2.1 ± 0.6 | 14 ± 1 | 0.055 |
| 4 | DA-CNF (4.8) | 4.5 | 143 ± 5 | 2.5 ± 0.5 | 8 ± 1 | 0.095 |

Source data are provided as a Source Data file.

**Table 2 | Optimization of reaction conditions for preparation of R-ON-CNF**

| Entry | DO (mmol/g) | Bn-ONH$_3^+$Cl$^-$, equiv. | DF (mmol/g) Weight | DF (mmol/g) $^{13}$C CP/MAS NMR | Conversion of aldehyde groups (gravimetrical, $^{13}$C CP/MAS NMR) (%) | Portion of incorporated Bn group from initial Bn-ONH$_3^+$Cl$^-$ (%) |
|---|---|---|---|---|---|---|
| 1 | 2.5 | 1 | 2 | 1.3 | 80, 52 | 80, 52 |
| 2 | 4.8 | 1.1 | 3 | 2.5 | 63, 52 | 57, 47 |
| 3 | 4.8 | 1.5 | 3.5 | 3.3 | 73, 69 | 49, 46 |
| 4 | 4.8 | 4 | 5 | 4.6 | >99, 96 | 25, 24 |

Reaction conditions: Room temperature, solvent: H$_2$O, pH = 4, $t$ = 12 h. DO—degree of oxidation (amount of aldehyde groups).

oxidation. The reduction in crystallinity of cellulose follows two different regimes with oxidation conversion, where the rate is lower at the beginning, followed by a sharp increase after 25% carbohydrate conversion. A possible interpretation is that the hemicelluloses are primarily oxidized first then, followed by cellulose oxidation.

Quasi-static mechanical properties of the dry films were assessed (Table 1). Interestingly, films with a higher degree of oxidation (and thus a lower degree of crystallinity) show a stepwise drop in tensile strength, influenced by inferior in-plane CNF orientation (see increased film thicknesses). Still, Young's modulus does not decrease until the oxidation becomes as high as 4.8 mmol/g. This is probably due to interfibrillar cross-linking and stiffening of the film via hemiacetal linkages formed between aldehyde groups and hydroxyl groups of neighboring cellulose fibrils[56].

### Orthogonal functionalization of DA-CNF to R-ON-CNF

Following the oxidation of Holo-CNF to form DA-CNF, we explored the orthogonal functionalization method via oximation towards substituted R-ON-CNF films. The optimization was performed by placing the DA-CNF wet cake into an aqueous system containing benzylhydroxyl amine hydrochloride (Bn-ONH$_3^+$ Cl$^-$) at pH 4. The reaction proceeded at ambient conditions, and progress was monitored by the pH change of the solution (as the reaction proceeds pH drops due to the release of HCl). As the reaction was completed, the modified benzyl-ON-CNF (Bn-ON-CNF) wet cake was removed from the solution and washed with water. The film was then dried under reduced pressure at 93 °C to form a Bn-ON-CNF film. If the reaction is performed at different equivalents of reactant (Bn-ONH$_3^+$ Cl$^-$) or content of aldehydes in the DA-CNF wet cakes, the degree of functionalization (DF—mmol of installed functional group per gram of DA-CNF) of the final material can be tailored (Table 2). The DF was assessed both gravimetrically (by weighting a dry sample after the reaction) and by solid-state NMR ($^{13}$C CP/MAS NMR).

The DF gave comparable results determined gravimetrically or through solid-state NMR, with slightly lower values by NMR (Table 2). When 1.5 equiv. of Bn-ONH$_3^+$Cl$^-$ is used, 73% conversion of aldehyde groups is obtained after 12 h at ambient temperature, and near quantitative conversion with 4 equiv. at the same reaction conditions. Note that the R-ON-CNF wet cake is removed from the reaction mixture, allowing easy recycling of unreacted salts of O-substituted hydroxyl amines in subsequent reaction runs after

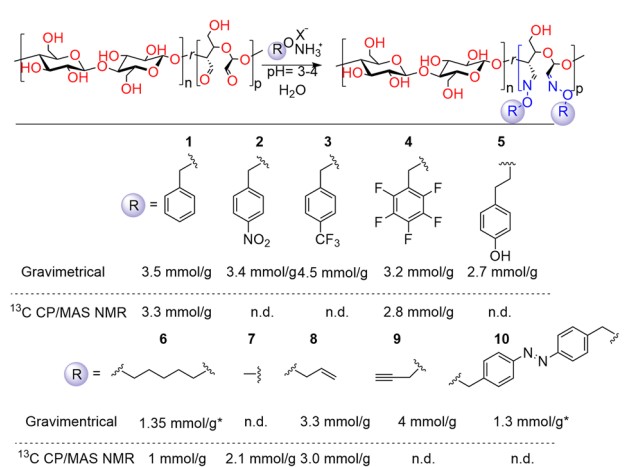

**Fig. 3 | Scope of O-substituted hydroxyl amines.** Reaction conditions: ca. 150 mg of DA-CNF (dry basis), 0.12 M solution of RONH$_3^+$ X$^-$ (~1.5 equiv.), 10 mL H$_2$O, room temperature, overnight; X = Cl or OTf. Reaction is performed on DA-CNF with 4.8 mmol/g of aldehyde groups. *The reactions are performed on DA-CNF film with 3.2 mmol/g of aldehyde groups.

adjusting the pH. Overall, the reaction design allows for a scalable and straightforward protocol for modifying CNF at ambient temperature with water as the only reaction medium. With the optimized conditions in hand (entry 3, Table 2) we screened a broad range of O-substituted hydroxyl amines with different functionalities. Under these reaction conditions, we covalently installed methyl, allyl, propargyl, benzyl, p-trifluoromethyl benzyl, pentafluorobenzyl, phenolic, and p-nitro benzyl, as well as difunctional, pentyl and azobenzenyl groups into the CNF network (Fig. 3). The corresponding salts of O-substituted hydroxylamines (apart of commercially available Bn-ONH$_3^+$ Cl$^-$ and Me-ONH$_3^+$ Cl$^-$) were synthesized from hydroxylamine and corresponding bromides[57], while synthesis from alcohols is also possible[58]. In the case of compounds **1–5, 8–10**, the R-ON-CNF wet cake stiffens during the progression of the reaction. In contrast, Me- and pentyl-functionalized DA-CNF wet cakes stayed soft and more flexible.

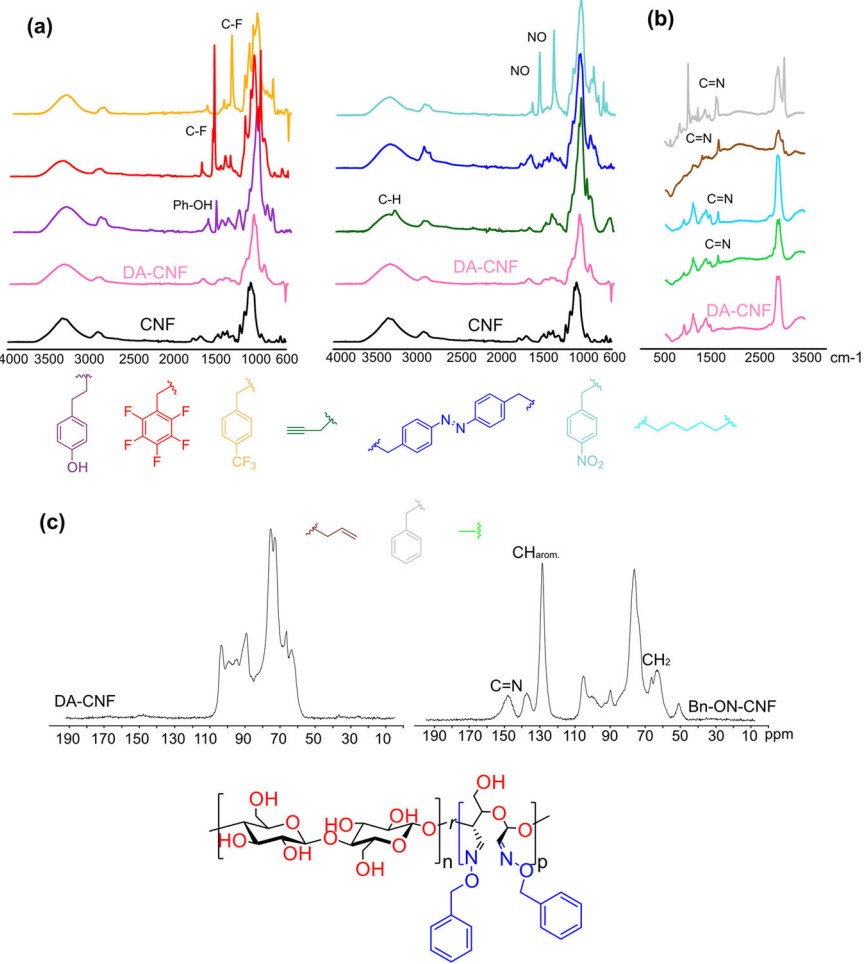

**Fig. 4 | Characterization of R-ON-CNFs. a** FTIR spectra of native (CNF), oxidized (DA-CNF), and modified (R-ON-CNF) films containing the following substituents: phenol (purple), pentafluorobenzyl (red), *p*-CF₃ benzyl (orange), propargyl (dark green), azobenzyl (blue), *p*-NO₂ benzyl (turquoise), pentyl (light blue), allyl (brown), benzyl (gray), methyl (light green). **b** Raman spectra of DA-CNF and selected R-ON-CNF films. **c** $^{13}$C CP/MAS NMR spectra of DA-CNF (left) and Bn-ON-CNF (right) as a representative example.

For monofunctional hydroxylamines, the amount of introduced functional groups was in a range of 2.7–4.5 mmol/g of DA-CNF, using DA-CNF with an aldehyde content of 4.8 mmol/g as the starting substrate and a slight access (1.5 equiv.) of the corresponding ether of hydroxyl amine (corresponding to 56–73% conversion of aldehyde groups). In the case of difunctional hydroxylamines, the reaction was performed on a DA-CNF starting substrate with a lower content of aldehydes, 3.2 mmol/g (to avoid a high degree of cross-linking leading to the formation of brittle samples). Since each difunctional hydroxylamine potentially reacts with two aldehydes, the total amount of substituent is lower (**6** and **10**). Thus, 1.3 mmol/g of introduced substituents corresponds to 2.6 mmol/g of reacted aldehyde groups.

To further increase the sustainability of the process, bio-based aromatic compounds (e.g., vanillin or furfural-based) can be considered. In addition, for substrate scope studies the reaction was driven to high DF to facilitate the characterization of the final products. However, the designed methodology allows for simple modulation of DF, where lower DF (mainly surface modification of nanofibrils) is desirable when the mechanical properties of the material are of the greatest importance.

## Chemical characterization of DA-CNF and R-ON-CNF films

Due to the poorly defined surface structure of CNF, the characterization of the modified CNF is challenging[59]. Thus, a combination of several spectroscopic methods is required to fully elucidate the reaction outcome. We first characterized DA-CNF films using FTIR spectroscopy (Fig. 4a). A successful oxidation was evident by an increased intensity in the region corresponding to C−O bond vibrations (880–890 cm⁻¹), which is due to the formation of acetals and hemiacetals between installed aldehyde groups and hydroxyl groups of cellulose[60,61]. An absence of the signal corresponding to C=O group of dialdehyde cellulose due to the formation of hemiacetal/acetal linkages was well documented in previous studies[62–64]. Successful functionalization of substrates containing highly polar groups was also confirmed by FTIR; *p*-trifluoromethyl benzyl (CF₃Bn)−1324 cm⁻¹ (C−F), pentafluorobenzyl (PFB)−1324 cm⁻¹ (Ar−F), *p*-nitro benzyl (NB)−1521 cm⁻¹ and 1343 cm⁻¹ (N−O sym. and asym.), phenolic−1200–1300 cm⁻¹ (Ar−O), propargyl−3290 cm⁻¹ (C−H) (Fig. 4a)[65].

Due to the overlap of methyl (Me), pentyl, allyl, and benzyl (Bn) substituents with cellulose signals, these films were analyzed using Raman spectroscopy. We obtained a clear indication of the successful derivatization by observing a distinct signal at around 1660 cm⁻¹ for all above mentioned R-ON-CNF films. This signal corresponds to C=N bond of the formed oximes[65] and is absent in the Raman spectrum of DA-CNF (Fig. 4b). To further validate FTIR and Raman results, we performed solid-state NMR ($^{13}$C CP/MAS) analysis on the DA-CNF and R-ON-CNF films; see Fig. 4c as a representative example with DA-CNF and Bn-ON-CNF (for more examples, see Supplementary Figs. 26–34). The solid-state NMR spectrum of DA-CNF correlates with previous reports, showing both acetal and hemiacetal formation of the

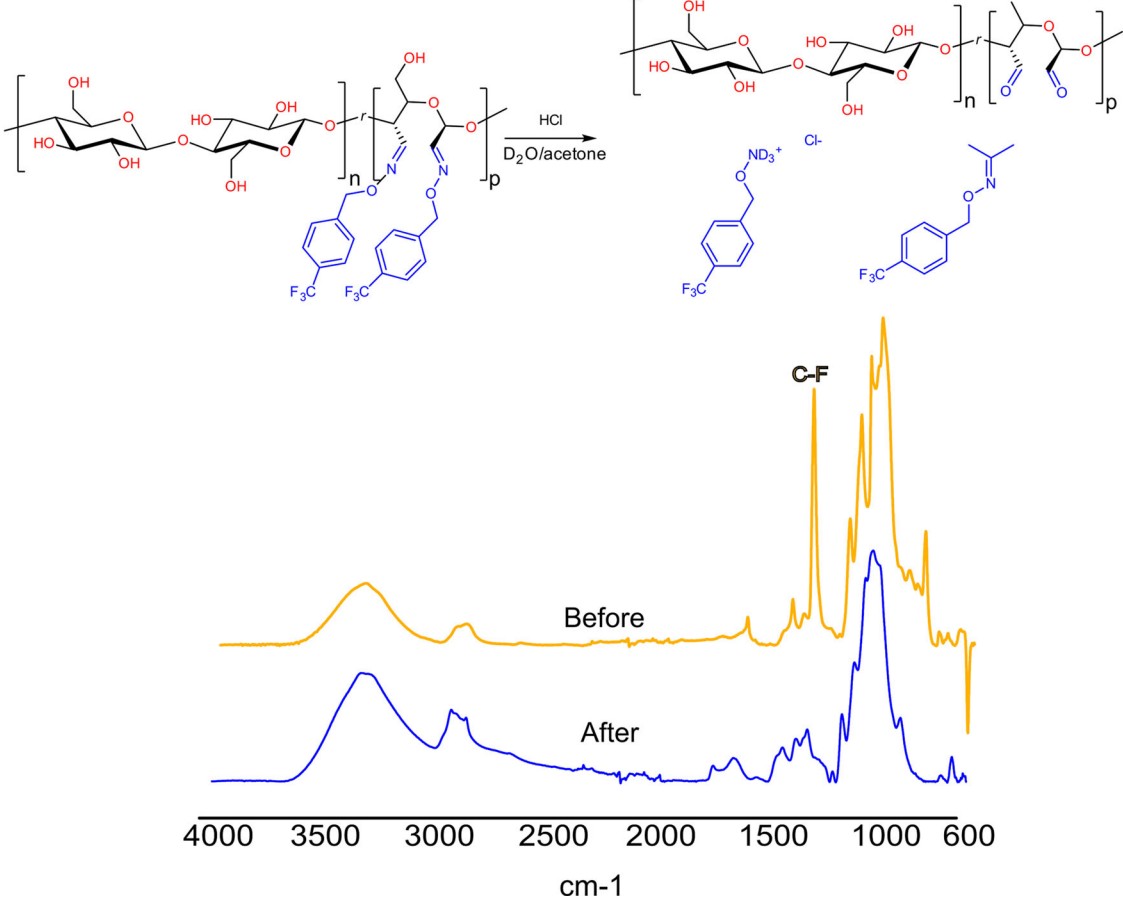

**Fig. 5 | Defunctionalization of R-ON-CNF.** Slow release of the functional group (CF$_3$Bn) from the CF$_3$Bn-ON-CNF in 1 M HCl solution in acetone/water (1/1, v/v) at room temperature; FTIR spectra of CF$_3$Bn-ON-CNF film in 1 M HCl solution in acetone/water (1/1, v/v) at 55 °C (before (orange) and after (blue)).

dialdehydes at around 90–100 ppm[61]. In the modified film, Bn-ON-CNF, the signal intensity at 90–100 ppm decreases, and additional signals of the aromatic ring (120–140 ppm) and oxime linkage (150 ppm) appear. However, the benzylic carbon (CH$_2$) signal overlaps with cellulose in the 70–80 ppm region. Integration of the signal corresponding to the oxime carbon atom (150 ppm) and carbon atoms of cellulose to estimate the DF (for details, see Supplementary Fig. 1) resulted in DF values that were in good agreement with gravimetrical estimation.

## Stability and defunctionalization of R-ON-CNF films

The possibility to selectively remove functional groups on cellulose nanofibers is of great importance, allowing for the recovery of the functional moiety and recycling of CNF. It is important to mention that upon the defunctionalization DA-CNF is recovered, rather than the initial Holo-CNF. However, several studies demonstrated that DA-CNF is also biodegradable, meaning that the post-life treatment of defunctionalized DA-CNF is significantly easier compared to the functionalized CNF material[66,67]. The stability and detachment conditions of the R-ON-CNF-modified films were explored with the CF$_3$Bn-ON-CNF due to the possibility to track changes by [19]F NMR. CF$_3$Bn-ON-CNF film was placed into 2 M solutions of HCl in water at ambient conditions for 24 h, followed by washing with water and drying under reduced pressure. Under these conditions, no apparent weight loss (<1 wt%) was observed. Even though partial depolymerization of cellulose may occur under aqueous acidic conditions, due to the hydrophobic nature of the film penetration of the solution into the films is minimized. On the contrary, when the film was placed in the mixture of acetone/D$_2$O, containing HCl (1 M), a slow release of the CF$_3$Bn-

ONH$_3$$^+$Cl$^-$ and the product of its reaction with acetone (Fig. 5, Supplementary Fig. 2) was detected by [19]F NMR. Using pentafluorobenzyl bromide as an internal standard enabled us to monitor the detachment kinetics. The complete detachment was achieved after 54 h according to [19]F NMR. However, after washing, FTIR showed the presence of a residual C–F signal. To achieve full disappearance of the signal, films were treated at 55 °C for 24 h in an acetone/water solution (1/1, v/v), containing HCl (1 M) (Fig. 5, Supplementary Figs. 35 and 36). The defunctionalized films appeared colorless when the reaction was performed in an acetone/water mixture at room temperature. Slight coloration occurred when the film was further treated under heating for the complete removal of the functional group.

## R-ON-CNF films with tailored, optical, hydrophobic, and mechanical properties

Functionalization of the CNF network can contribute to the tailoring of optical, mechanical, and functional properties of CNF materials. With the library of R-ON-CNF films in hand, we explored several potential applications of the prepared R-ON-CNF. Tailoring the optical properties of CNF films is highly interesting in application in "green" electronics and solar cell devices[68,69]. Among them, films possessing high visible light transmittance while lowered transmittance in the ultraviolet (UV) region are of interest. Reduced transmittance in the UV region can relate to increased light scattering (especially isotropic Rayleigh scattering on nanosized defects), which compromises transmittance in the visible range (especially pronounced for thicker samples) or installation of functional groups that absorb light in the UV region. Thus, while unmodified 0.033 mm-thick Holo-CNF film demonstrates reduced transmittance in the UV region (24% at 300 nm,

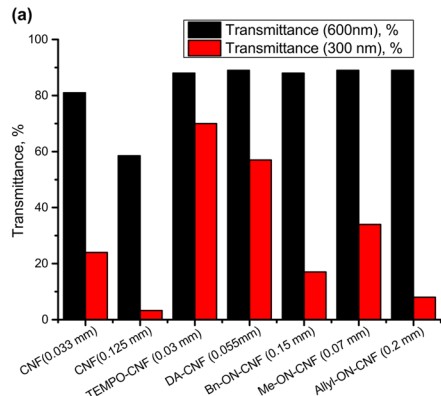

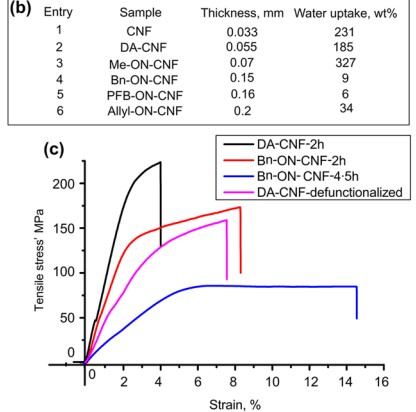

**Fig. 6 | Properties of CNF, DA-CNF, and R-ON-CNF. a** Optical properties of DA-CNF (3.2 mmol/g of aldehyde groups) and R-ON-CNF, the transmittance at 600 nm (black) and 300 nm (red), Supplementary Figs. 3 and 4. **b** Water uptake of CNF, DA-CNF, and R-ON-CNF films. **c** Tensile properties of DA-CNF-2h (black), Bn-ON-CNF-2h (red), and Bn-ON-CNF-4.5 h (blue) (oxidized for 2 and 4.5 h, with aldehyde content of 2.6 and 4.8 mmol/g, respectively) and defunctionalized DA-CNF films (magenta). Source data are provided as a Source Data file.

Fig. 6a), transmittance in the visible range is also reduced (81% at 600 nm, Fig. 6a). This becomes even more pronounced when the thickness of the film increases to 0.125 mm, where transmittance at 600 nm decreases to 58% (Fig. 6a). In contrast, modified R-ON-CNF films show very high transmittance in the visible range (88% at 600 nm, Fig. 6a), most likely because fibrils are highly individualized due to the surface functionalization. Reduced transmittance in the UV region (8–34% at 300 nm, Fig. 6a) is caused by the presence of absorbing oxime moiety (Me-ON-CNF) and aromatic (Bn-ON-CNF, PFB-ON-CNF) or allylic substituents (Allyl-ON-CNF). These results indicate that the developed method allows the formation of thick films with UV-blocking properties while maintaining high transmittance in the visible light range.

In addition, due to a more plastic structure of oxidized DA-CNF and modified R-ON-CNF films compared to native CNF we were able to modulate their haze by changing the roughness of the films' surface. The change in the haze was from 9% to 50% for DA-CNF, from 22% to 60% for Bn-ON-CNF, and from 11% to 80% for Me-ON-CNF (Supplementary Fig. 5, Supplementary Table 3). This can govern the final application of the materials, e.g., high transmittance/low haze materials are of interest, e.g., indoor displays, while high transmittance and high haze are desired for LED electronic devices applications[69]. The present approach also allowed us to control hydrophilicity through surface modification of cellulose nanofibers, which is one of the main shortcomings of CNF films limiting their applications (Fig. 6b).

A slight drop in water uptake is seen for DA-CNF films compared to native CNF, which is in line with acetal formation. There was a significant drop in water uptake for Allyl-ON-CNF (from 185% to 34%) and even more pronounced for the bulkier and more hydrophobic substituents, Bn-ON-CNF and PFB-ON-CNF (to 9% and 6%, respectively). In contrast, Me-ON-CNF demonstrated higher water uptake (327%), similar to methylated cellulose, which is known to be soluble in cold water[70].

The mechanical properties of R-ON-CNF were studied in detail for Bn-ON-CNF films prepared from DA-CNF oxidized for 2 h (ca. 2.6 mmol/g of aldehyde groups) and for 4.5 h (ca. 4.8 mmol/g of aldehyde groups). Representative strain–stress curves for DA-CNF and Bn-ON-CNF films are presented in Fig. 6c. The material becomes more ductile with an increased degree of functionalization. The increased ductility of Bn-ON-CNF-2h compared to unfunctionalized DA-CNF-2h, with the same oxidation degree, is partly due to reduced cross-linking via intramolecular acetals due to the formation of oxime linkages, and due to the change of the intrinsic properties of the CNF fibrils, making them more prone to plastic deformation. The ductility increased even

further for Bn-ON-CNF-4.5 h to give elongation at a break of ca. 14%. The decreased number of acetal cross-linkages in R-ON-CNF is also responsible for lower tensile strength for highly functionalized DA-CNF films. This shows that the designed method allows for simple tuning of the final material's properties toward targeted applications. We also tested DA-CNF films obtained after the defunctionalization of Bn-ON-CNF-2h films. The films were soaked in acetone/water (1/1, v/v) HCl (1.2 M) solutions for 36 h at room temperature to detach Bn groups, washed with acetone and water, and dried at 93 °C under vacuum. Only a slight drop in tensile strength was observed compared with the reference (Fig. 6c), showing that upon the defunctionalization the structure of DA-CNF can be preserved to a large extent.

### Targeted applications—CNF-based triboelectric generators

As a demonstration of the versatility of the functionalization platform in targeted applications, we evaluated functionalized films based on DA-CNF nanofibers of different chemical surface functionalities for triboelectric nanogenerators (TENG)[71]. TENGs consist of two dielectric materials with different affinities to accept or donate electrons. When these two materials are assembled as a TENG, and brought into contact periodically, the electrostatic charges are generated on the surfaces due to contact electrification. The build-up of electrostatic charges on the surfaces results in an increased electrostatic field. Upon separation of two surfaces, a potential drop is created between opposite electrodes, and to balance it electrons flow from one to another electrode via an external load. Further, when the surfaces are again contacted, the potential drop disappears, and electrons flow back[72].

The most common tribo-negative materials used in TENGs are polytetrafluoroethylene (PTFE), fluorinated ethylene propylene (FEP), or polypropylene (PP), while for tribo-positive materials it is polyvinyl alcohol (PVA), polyvinyl acetate (PVAc), nylons, etc. Environmental concerns regarding these petroleum-based and difficult-to-degrade materials raised interest in the development of bio-based counterparts. Recently a high potential of cellulose-based materials for fabrication of TENGs was discovered and resulted in the development of TENG devices from bacterial cellulose[73], nitro and methylated CNF[74], regenerated cellulose[75], etc.

Inspired by this, we envisioned the developed reversible functionalization methodology as a simple tool for tuning triboelectric properties of CNF, bringing the development of TENGs closer to their synthetic polymer counterparts. We used the following surface functionalities in the CNF films: *p*-nitro benzyl (NB), pentafluorobenzyl (PFB) for tribo-negative materials, and methyl (Me) and acetal/hemiacetal (DA-CNF) for tribo-positive materials. Modification of CNF with

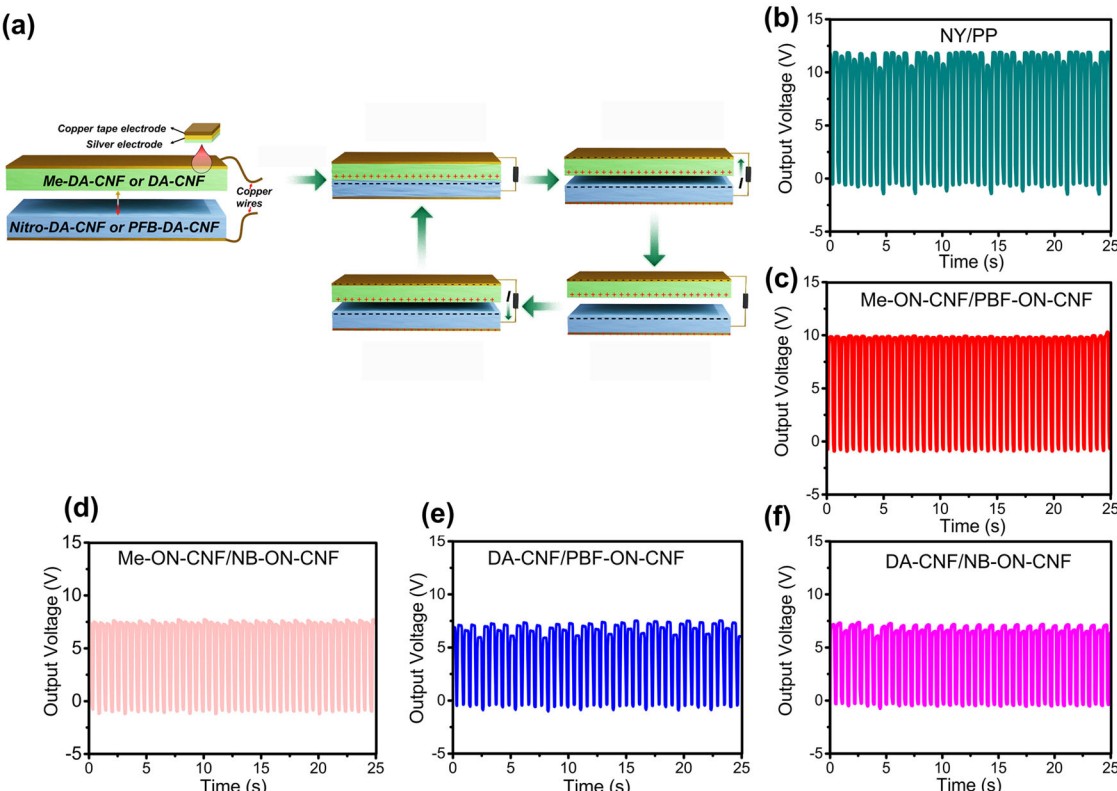

**Fig. 7 | Triboelectric properties of DA-CNF and R-ON-DA-CNF films. a** Working mechanism of TENG. **b** Output voltage of PP/NY pair. **c** Output voltage of Me-ON-CNF/PBF-ON-CNF pair. **d** Output voltage of Me-ON-CNF/NB-ON-CNF. **e** Output voltage of DA-CNF/PFB-ON-CNF. **f** Output voltage of DA-CNF/NB-ON-CNF.

small organic molecules rather than with polymers coupled with selective defunctionalization contributes to the circularity and environmental amiability of these materials in comparison to chemically inert and difficult-to-degrade polymers generally used for TENGs. Figure 7a shows the working principle behind the device. Figure 7c−f demonstrates the output voltage obtained for the pairs between Me-ON-CNF/NB-ON-CNF, Me-ON-CNF/PFB-ON-CNF, and DA-CNF/NB-ON-CNF and DA-CNF/PFB-ON-CNF, compared to the reference pair−polyamide-11 (NY)/polypropylene (PP) (Fig. 7b). The highest output voltage was achieved for Me-DA-CNF/PFB-ON-CNF pair, which constituted around 83% of the reference pair (10 V). It should be noted that these numbers can be further improved by increasing the surface area and roughness of the films[74].

## Discussion

High-end applications of CNF often require tailored surface chemistry, which is achieved via the chemical modification of CNF. The challenge is to carry out the transformation on a highly hydrophilic substrate (CNF) while the vast majority of organic transformations are incompatible with water. Moreover, to maintain the green aspect of the final CNF-based material and address the post-life treatment it is highly desirable to be able to defunctionalize CNF on demand. In a search for a CNF modification platform fulfilling the following criteria: (i) insensitivity to water, (ii) generality, and (iii) on-demand reversibility. Functionalization of cellulose nanofibers rather than cellulose fibers is beneficial due to the higher accessibility of the cellulose chains and more homogeneous distribution of the functional groups in the final material. We envisioned a two-step process where CNF is first oxidized to dialdehyde CNF and then reacted with salts of O-substituted hydroxyl amines. In order to avoid coagulation of CNF in the solution of the oxidant and simplify the isolation of the product we envisioned performing NaIO$_4$ oxidation of preformed Holo-CNF wet cakes. We proposed that subsequent modification of

the installed aldehyde groups via oximes, which are known to possess higher hydrolytic stability compared to imines, will allow for achieving relatively high stability of the linkage keeping the possibility of detachment in the presence of a trapping agent. We found the transformation to be highly general, as demonstrated by the covalent attachment of substituted benzylic, allylic, propargylic, aliphatic, and phenolic substituents. The degree of functionalization is easily tuned by the reaction time and/or reactant concentration during the oxidation or functionalization step, enabling controlled surface modification of CNF with a maintained internal nanostructure. Still, in the extreme case, we could reach a degree of functionalization as high as 4.6 mmol/g. The complete detachment was achieved with acetone as a trapping agent at ambient temperature, allowing for recovery of both DA-CNF and the corresponding functional moiety. To study the feasibility of the protocol for the production of materials we prepared R-ON-CNF films. The films possess very high optical transmittance in the visible light range (higher than native CNF) with decreased transmittance in the UV region from functional groups and tunable haze, advocating their potential in photonics. In addition, the R-ON-CNF was explored in electronic devices in terms of triboelectric properties. The methodology developed addresses challenges in efficient and sustainable manufacturing and post-life treatment of CNF-based materials in targeted applications.

## Methods

### Material preparation

#### Preparation of *O*-substituted hydroxyl amines

**Synthesis of *O*-allylhydroxylamine(8) and *O*-propargylhydroxylamine hydrochlorides (9).** *O*-allylhydroxylamine and *O*-propargylhydroxylamine hydrochlorides were prepared as follows. Hydroxylamine hydrochloride (1.1 g, 15.9 mmol) was dissolved in 16 mL EtOAc/H$_2$O (1/2, v/v) mixture, containing NaOH (1.36 g, 34 mmol). The

solution was stirred at 0 °C for 4 h. Allyl bromide (1.2 mL, 13.9 mmol) or propargyl bromide (1.2 mL, 13.9 mmol) was added and the reaction mixture was stirred at 50 °C for several hours. The reaction mixture was extracted with EtOAc/H$_2$O. The combined organic phase was collected and dried over anhydrous Na$_2$SO$_4$. The solvent was removed under reduced pressure using a rotary evaporator. The obtained oil was dissolved in EtOH (100 mL) and 0.150 mL of HCl (conc.) was added to the solution. The solution was stirred at 65 °C for 6 h. Upon the completion of the reaction, solvent was removed under reduced pressure using a rotary evaporator to give the products yellowish solids.

**General procedure for preparation O-substituted hydroxylamine hydrotriflates (2, 3, 4, 5, 10).** Boc-protected hydroxylamine (1.05 equiv., for bifunctional bromides 2.1 equiv. were used), corresponding bromide (1 equiv.), 1,8-Diazabicyclo(5.4.0)undec-7-ene (DBU) (1.05 equiv., for bifunctional bromides 2.1 equiv. were used) were dissolved in DCM. The reaction mixture was stirred at room temperature for several hours. The progress of the reaction was monitored by NMR. Upon completion, the reaction mixture was extracted with water. The combined organic phase was collected and dried over anhydrous Na$_2$SO$_4$. The obtained DCM solution was cooled to 0 °C, and triflic acid (1 equiv.) was added to the solution dropwise. The reaction mixture was left to stir for an hour. The reaction was accompanied by an evolution of gas and the formation of a precipitate. The final product was collected by vacuum filtration, washed with cold DCM, and dried under reduced pressure.

**O,O'-bis-(1,5-dihydroxylamine)pentane hydrotriflate (6).** The reaction was performed in THF at 80 °C. The progress of the reaction was monitored by NMR. An additional amount of DBU (ca. 0.3 equiv.) was added to the reaction mixture after ca. 6 h. The final product (1,5-dihydroxylaminepentane hydrotriflate) did not form a precipitate upon the addition of triflic acid. The solvent was removed under reduced pressure and the flask containing the product was left under vacuum at room temperature overnight. The developed white crystals were collected and washed with hexane.

**Preparation of Holo-CNF wet cake.** Holo-CNF was prepared according to the previously published procedure[5]. Briefly, holocellulose fibers were obtained using a mild peracetic acid (PAA) delignification on Picea abies spruce. Spruce pieces were treated with PAA (4 wt% in water, pH was adjusted to 4.8 with NaOH before reaction) for 45 min at 85 °C. Four rounds of PAA treatments were performed. The final fibers were washed with 0.01 M NaOH and then extensively with deionized water to remove residual chemicals. Then they were subjected to mechanical defibrillation using a high-pressure microfluidizer (Microfluidizer M-110EH, Microfluidics Corp., USA). The homogenization process was conducted employing two serial coupled Z-shaped interaction chambers with path diameters of 400 and 200 μm at a pressure of about 600 bar for two passes, followed by employing two serial coupled Z-shaped interaction chambers with path diameters of 200 and 100 μm at a pressure of about 1500 bar for three passes.

A suspension of Holo-CNF in water (ca. 0.6 wt%) was diluted to ca. 0.04 wt% with Milli-Q water. The wet cake was prepared via vacuum filtration using Durapore® Membrane Filter, 0.22 μm (VWR) to give the wet cake with ca. 0.25 g of CNF (dry basis).

**Preparation of DA-CNF wet cake.** A CNF wet cake obtained as described above (ca. 0.25 g of CNF on a dry basis) was carefully transferred into the beaker containing a solution of NaIO$_4$ in water (40 mL). The beaker was covered with aluminum foil to prevent the reaction mixture from light exposure. Upon the completion of the reaction obtained DA-CNF wet cake was removed from the beaker,

containing NaIO$_4$, and transferred into another beaker containing ca. 800 mL of distilled water and left to soak, followed by washing with distilled water. To assure the complete removal of NaIO$_4$ from the wet cake a piece of wet cake can be placed into a solution of hydroxylamine hydrochloride (400 mg in 25 mL of water). In the case of NaIO$_4$ presence, the wet cake develops a yellow coloration due to a generation of iodine.

**Preparation of R-ON-CNF wet cake.** A DA-CNF wet cake (ca. 150–200 mg of DA-CNF on a dry basis) prepared as described above was placed into a water solution (10 mL) of the corresponding O-substituted hydroxylamine hydrochloride or hydrotriflate. The pH was adjusted to 4 using a solution of NaOH. The reaction was performed at room temperature overnight. Upon the completion of the reaction, the obtained R-ON-CNF wet cake was removed from the solution and placed into a beaker containing ca. 800 mL of distilled water and left to soak, followed by washing with distilled water, MeCN, and distilled water again. p-Nitrobenzylhydroxylamine hydrotriflate and 4,4'-diaminoxymethylazobenzene hydrotriflate showed incomplete solubility in water at the initial pH, however, solubility improved during the course of the reaction due to the drop of the pH. In the case of 4,4'-diaminoxymethylazobenzene hydrotriflate small amount of MeCN (2.5 mL) was added to improve the initial solubility.

For substrate scope studies the reactions were performed using ca. 1.5 equiv. of the corresponding salt of O-substituted hydroxylamine, using DA-CNF wet cakes with the content of aldehyde groups of 4.8 mmol/g (for mono-functional O-substituted hydroxylamine salts) and of 3.2 mmol/g (for bifunctional O-substituted hydroxylamine salts).

**Preparation of dry films from CNF, DA-CNF, and R-ON-CNF wet cakes.** Dry films of CNF, DA-CNF, and R-ON-CNF were prepared using Rapid Köthen under vacuum (ca. 70 mbar) at 93 °C for 20 min. A wet cake was "sandwiched" in between either two Teflon sheets (for films with smooth surfaces) or in between two Durapore® Membrane Filters, 0.22 μm (VWR). The obtained "sandwich" was in turn placed in between two thick paper sheets and subjected to drying in Rapid Köthen.

## Characterization

**Nuclear magnetic resonance.** [1]H NMR spectra were recorded with a Bruker 400 (400 MHz) spectrometer as solutions. Chemical shifts are expressed in parts per million (ppm, δ) and are referenced to a deuterated solvent as an internal standard. [13]C NMR spectra were recorded with a Bruker 400 (100 MHz) spectrometer as solution.

**Solid-state nuclear magnetic resonance.** The solid-state NMR measurements were performed on a Bruker Avance-HD 500 MHz spectrometer equipped with a high-power 4 mm magic-angle spinning (MAS) probe. Cross-polarization MAS (CP-MAS) [13]C spectra were recorded at a resonance frequency of 125.7 MHz and a sample spinning speed of 10 kHz. For the ramped CP, the radio-frequency fields with average nutation frequencies of 80 kHz and 1–2 ms contact time were used. Proton heteronuclear decoupling was achieved by TPPM pulse sequence at a nutation frequency of 80 kHz[76]. Typically, 6k transients were acquired with the recycling delay of 8 s. Sample temperature was stabilized to 25 °C. Carbon-13 chemical shifts were referenced externally to the CH carbon of solid adamantane at 29.5 ppm[77]. For the quantitative analysis of CP-MAS spectra, it was verified that the relative intensities of the spectral lines do not significantly change with a further increase of the contact time up to the limit where relaxation damping of spin-locked magnetization becomes effective. Additionally, linear variation of the radio-frequency field strength ("ramp") makes spectra not sensitive to exact Hartman-Hahn field matching for different functional groups.

**Fourier transform infrared.** FTIR spectra were recorded using a Spectrum 100 Fourier transform infrared (FTIR) spectrometer (PerkinElmer, USA) equipped with a Golden Gate diamond ATR (Gaseby Specac Ltd, UK).

**Microscopy.** The SEM images were obtained using a field-emission scanning electron microscope (FE-SEM, Hitachi S-4800, Japan). All samples were Pt/Pd sputtered under vacuum (Cressington R208, UK).

Energy dispersive X-ray spectroscopy (EDS) mapping was performed at an acceleration voltage of 5 kV using FE-SEM equipped with an X-Max 80 mm$^2$ silicon drift detector (Oxford Instruments, UK).

**Tensile test.** The tensile properties of the samples were determined using a universal testing machine (Instron 5944, USA) equipped with a 2 kN load cell and a video extensometer. The samples (generally 2.5 mm × 0.5 cm) were preconditioned for 48 h at 50% RH before measurement. The tensile properties tests were carried out at a temperature of 22 °C and 50% RH, with a 10% min$^{-1}$ strain rate and.

**Optical properties.** Optical transmittance of the samples was acquired using UV-2550 Shimadzu spectrophotometer from 200–800 nm, equipped with an integrating sphere consisting of a hollow spherical cavity covered with $BaSO_4$ reflective coating (to measure total transmittance, $T_{total}$) and not covered (to measure diffused transmittance, $T_{diffused}$). Haze was calculated as follows:

$$\text{Haze}(\%) = \frac{T_{diffused}}{T_{total}} \qquad (1)$$

where $T_{diffused}$ is a diffused transmittance, and $T_{total}$ is a total transmittance.

**X-ray powder diffraction (XRD).** XRD was performed using a Thermo Fisher Scientific ARL X'TRA powder diffractometer through CuKα radiation at 40 mA and 45 kV. The scans were performed over $2\theta$ of 5–50° with a step size of 0.04°. The crystallinity index (CI) was obtained as follows:

$$\text{CI} = \frac{I_{200} - I_{am}}{I_{200}} \qquad (2)$$

where $I_{200}$ is a total intensity of [200] peak, and $I_{200} - I_{am}$ is an intensity of crystalline [200] peak.

**Raman spectroscopy**
Raman spectra of CNF films were measured with a confocal Raman microscope (Jobin Yvon HR800 UV, Horiba) using a 514 nm laser (Stellar-Pro, Modu-laser).

## Data availability
The datasets generated and/or analyzed during the current study are supplied in the supplementary information. If additional data or information is sought, this will be provided by the corresponding authors upon request. Source data are provided with this paper.

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

## Acknowledgements
P.O. acknowledges Formas—a Swedish Research Council for Sustainable Develop-ment (Re-Design Plastic, 2020-01696).

## Author contributions
E.S. conceived the idea, performed the synthesis of the materials and their characterization, wrote the manuscript and supporting information. F.R. fabricated and characterized TENG devices and participated in manuscript preparation. S.V.D. performed all solid-state NMR experiments and participated in manuscript preparation. P.O. did project design with E.S., manuscript preparation, and funding acquisition. L.B. did the supervision, manuscript preparation, and funding acquisition.

## Funding

## Competing interests
The authors declare no competing interests.
