## [Peer Review File · Nature Communications]

Aqueous synthesis of highly functional, hydrophobic, and chemically recyclable cellulose nanomaterials through oxime ligationReviewers' comments:

Reviewer #1 (Remarks to the Author):

In this work, the reversible formation of oximes is proposed as a new functionalization of cellulose nanofibers. Though end group functionalization of cellulose and nanocellulose is a pretty well-known topic, the authors investigate several surface functionalizations by using different commercial or noncommercial hydroxylamines and demonstrate the generality of the proposed reaction approach, its viability and effectiveness. I think that from the "chemical" point of view the work is sound and the results are clear. In my opinion, the work is interesting but needs revision to meet the requirements of novelty and urgency to be published in Nature Communications.

- First of all, hydroxylamine binding is a method for titrating the available aldehyde groups. This somehow decreases the novelty of the work. Then, the introduction of dialdehyde groups onto surface anhydro glucose units in cellulose nanofibers is not new too. This work makes an excellent presentation of several surface functionalization of CNFs by the proposed reaction method. Further discussion is needed to support the true advantage of the reaction with hydroxylamines: are they effective where other methods failed?

- Relying on NaIO₄ oxidation, the authors introduce a high amount of aldehyde groups on the CNFs.
~4.5 mmol per g of CNF means 4.5 mmol aldehyde groups/6.2 mmol of cellulose are there, so 36% of AGU units have been oxidized. It is quite a high degree of oxidation. CNFs original structure is not preserved. I do not understand the usefulness of such an extensive skeleton functionalization. In the introduction, the authors declare that the challenges identified in this work include: "preserving the nanostructure of CNF, enabling high accessibility of the functional groups, and overcoming moisture sensitivity of the transformation". Indeed, it is accepted that a low degree of substitution/functionalization (DS<0.37) will preserve the nanomorphology of nanocellulose (D.-Y. Kim, Y. Nishiyama, and S. Kuga, "Surface acetylation of bacterial cellulose," Cellulose, vol. 9, no. 3-4, pp. 361-367, 2002). I think the authors should discuss more and perform the necessary characterizations also on samples with lower oxidation extent, but more preserved crystallinity.

- Microscopy investigation of the functionalized samples needs to be performed, either by AFM or scanning electron microscopy.

- Reversibility: how is the appearance of fibers after the removal of oximes?

- How do the authors know that CNFs are composed of 18 cellulose chains? Are there experimental data to support this statement?

- The first scheme of supporting information is wrong: the correct product is hydroxylamine hydrochloride, not amine hydrochloride.

- Please explain the following sentence: "In the case of more rigid hydrophobic substituents (e.g., R1-R5, R10), the R-ON-CNF wet cake stiffens during the progression of the reaction. In contrast, Me- and pentyl-functionalized DA-CNF wet cakes stayed soft and flexible."

- "Whether the difunctional hydroxylamine reacted with one or two aldehydes was assessed by using triflates as a counterion instead of halides." It would be much more reliable to look at the bifunctional reagents by XPS technique.

- The authors decided to study kinetics of oxime removal following CF₃-phenyl group dissolution by NMR spectroscopy. They also report an example with a pentafluorobenzene appendage on oximes. In my experience these rings are susceptible to nucleophile attack, did the authors check their water stability?

Reviewer #2 (Remarks to the Author):

This is an interesting paper, showing the preparation and characterization of DA-CNF and various R-ON-CNFs, and application of R-ON-CNFs to triboelectric generators, from some new aspects. Therefore, this manuscript can be publication in Nature Communications after revisions.

- 1) Since the R-ON-CNFs still contained some aldehyde groups, not only DA-CNF films but also R-ON-CNF films may be discolored after drying at 93C. This discoloration of aldehyde-containing films should be addressed in terms of aldehyde content.
- 2) It may be difficult in this study, but the yields (based on the starting dry CNF weight) and molecular weights of DA-CNFs prepared under various conditions should be measured or discussed. DA-CNFs may partly decrease in the yield by oxidation and washing processes.
- 3) Not only the conversion ratio from aldehyde groups to oximes (i.e., 73% at 1.5 eq. of Bn-ONH₃+Cl⁻ in page 5) but also the efficiencies of oxime formation (i.e., oximed amine/added amine by mol) should be added in Table 1).
- 4) It is better also to describe the reaction efficiency of NaIO₄ in the DA-CNF preparation. Namely, the molar ratio of NaIO₄ consumed for aldehyde formation/that added in the oxidation.
- 5) Page 7: The reaction conditions in acidic acetone/water at 55 degree C for 24 h may be harsh for R-ON-CNF. The low yield losses (<1%) were described in page 7, but some depolymerization of oxidized cellulose molecules may be unavoidable during de-oxime reactions, which should be addressed.
- 6) Figure 1: Since C.I decreased to almost zero after oxidation for 7.5 h, did this oxidation occur also in the crystalline inside CNFs without reaction selectively on the CNF surfaces?
- 7) Figure 2, green spectrum: Is it true that the C-H peak is located at ~3250 cm⁻¹?
- 8) Figure 3b: The changes in water uptakes were consistent with water contact angles on the films?
- 9) Figure 3a, S4, S5, and S6: It is better to include photographs of the films in this manuscript to show the colors or transparencies.
- 10) Figure S6: Why the films have such high haze values? Many pores with air/material interfaces are present in the films? What are the factors influencing the haze values of the films?
- 11) Figure S1: What is the signal at ~50 ppm owing to?
- 12) Figure S1 and Figure 2c: Since the signal pattern at 60-110 ppm in Figure S1 is similar to that in Figure 2C, this sample may have significant amounts of aldehyde groups (or the oximation ratio may be too low). Please explain this matter. Or it is better to show another C13-NMR spectrum of R-ON-CNF with a much higher degree of oximation.
- 13) Signal assignments should be added to Figure S3 and all 1H- and 13C-NMR spectra in Appendix. The solvents used in the 1H- and 13C-NMR measurement are also better to be described in the figures.

Reviewer #3 (Remarks to the Author):

The present manuscript with the title "In water synthesis of highly functional, hydrophobic, and chemically recyclable cellulose nanomaterials through oxime ligation" describes an interesting study on the chemical modification of nanostructured cellulosic materials. The manuscript is well structured, the work concisely described, the experimental procedures can be repeated based on the protocols and the illustrations are of satisfying quality.

Nevertheless, I believe that the work is not suited for publication in Nature Comm. There are two main reasons. First, the element of novelty, a prerequisite to publication, is largely missing. As the authors state themselves, periodate oxidation of carbohydrate (the very old Malaprade oxidation) is very well-known and has been extensively applied also to (nano)cellulosic materials. In most cases, those studies did not target the "dialdehyde cellulose" directly, but used it as an intermediate for further functionalization. Schiff bases as well as non-substituted and substituted hydrazones and oximes have been synthesized from dialdehyde cellulose and other dialdehyde polysaccharides. Despite placing the emphasis more on the properties of the products, the manuscript does not offer any novel aspects here.

Second, there are several aspects of the work which seem to indicate the studies have not yet fully matured (see below), and there are several claims which seem a bit exaggerated.

- line 3 (title): the products are certainly less hydrophilic than cellulose, but to call them hydrophobic would be quite an exaggeration (cf. for instance to hydrocarbons). The product still possesses a wealth of hydrophilic functions, and in case of the hydrochlorides or other salts are even ionic.
- line 76: not all modifications of CNF suffer from moisture sensitivity. There have been general approaches to reactions of CNF proceeding in the surface-confined water layer (Beaumont et al. Nature Comm. 2021) or at least being fully water-tolerant (Hettegger et al. Chem.Sus.Chem. 2015 and 2016).
- line 89: it is not fully clear what "prefunctionalization" means. TEMPO oxidation, periodate oxidation or acyl transfer catalyzed by imidazole would not require any preceding steps.
- line 89: the statement regarding atom economy would require a solid support by numbers to demonstrate that the proposed periodate oxidation / oxime approach is superior to the alternatives
- line 91: the statement of "harsh conditions" needed for functional group / substituent removal is not fully conclusive: a 0.1 M NaOH used for removal of acetyl/acyl groups is not harsher than the 1 M HCl used in the present work to cleave the oximes.
- line 105: this statement is not supported by a literature survey: the "dialdehyde" structure obtained by periodate oxidation is in very many cases used to have a reactive moiety for further modification (mostly by reacting with N- or C-nucleophiles). This further modification of the aldehydes is certainly not "less common".
- line 112: current reduction methods rather use aminoboranes which are selective towards imines and leave the aldehydes/hemiacetals unaffected.
- line 114: the evolution of hydrogen gas is strongly depending on the conditions and can be largely avoided by deeper temperatures, higher pH, alcoholic co-solvents, etc.
- line 122: it is correct that for oxime cleavage often ketone or aldehyde traps are used to scavenge the released hydroxylamine. However, these traps are bound to an appreciable extent by the dialdehyde polysaccharides. If this occurs by hemiacetal bonds, the reaction is reversible and the binding temporary, if covalent bonds are formed (acid-catalyzed aldol), the scavenger is consumed and the (surface) structure and chemistry of the polysaccharide irreversibly altered. This chemistry thus needs much further work to become fully reliable.
- line 143: the amount of acetal linkages is rather low. Moreover, if they exist they would not react in the proposed way with hydroxylamines.
- line 157: the use and fate of the "CNF wet-cake" is not fully clear. Why would it not disintegrate in the aqueous surroundings? Is the contact time with the periodate reagent (and the resulting extent of oxidation) everywhere in the cake the same?
- line 183: the problem with increasing iodate concentrations is also the increasing instability of periodate (of which the mechanism is unknown)
- lines 209-212: the reason for the selection of these substituents remains somewhat unclear. It appears as if just the commercially available O-substituted hydroxylamines were tested. This demonstrates the general applicability of the procedure, but would not support claims that there was any "targeted" selection.
- line 220/387 (e.g.): the reagents should be called O-substituted hydroxylamines, but not "ether of hydroxyl amine"
- lines 270/276/281-282: the "recycling of CNF" is quite an exaggeration. First, after cleavage, the product is not cellulose anymore, but periodate oxidized cellulose – the second modification step, the oxime formation, might be reversible, the first one, periodate oxidation, is not. Second, the conditions required for oxime cleavage (1M HCl, hydroxylamine scavengers), are rather detrimental to cellulose integrity. The acidity causes hydrolytic cleavage, which is already well

noticeable at that pH and reaction times, and the scavenger might react with the dialdehyde cellulose and modify it (see above).

- line 396: is such a high degree of oxidation / modification really compatible with the claim that only the surface of the CNF is modified?

- line 410: the experiments used holocellulose, which contains certain amounts of hemicelluloses which are more reactive than cellulose in periodate oxidations. This aspect is not considered, but it might imply that a certain amount of modified CNF is actually modified hemicellulose.

List of response to the reviewers' comments

Reviewer #1

In this work, the reversible formation of oximes is proposed as a new functionalization of cellulose nanofibers. Though end group functionalization of cellulose and nanocellulose is a pretty well-known topic, the authors investigate several surface functionalizations by using different commercial or noncommercial hydroxylamines and demonstrate the generality of the proposed reaction approach, its viability and effectiveness. I think that from the "chemical" point of view the work is sound and the results are clear. In my opinion, the work is interesting but needs revision to meet the requirements of novelty and urgency to be published in Nature Communications.

Comments:

1. First of all, hydroxylamine binding is a method for titrating the available aldehyde groups. This somehow decreases the novelty of the work. Then, the introduction of dialdehyde groups onto surface anhydro glucose units in cellulose nanofibers is not new too. This work makes an excellent presentation of several surface functionalization of CNFs by the proposed reaction method. Further discussion is needed to support the true advantage of the reaction with hydroxylamines: are they effective where other methods failed?

Response: We agree that the novelty and advantages of the developed method were not adequately highlighted in the old version of the manuscript. Indeed, oxidation of cellulose with NaIO_4 is a known and powerful method to perform the functionalization of cellulose fibers with aldehyde groups, opening vast opportunities for further modifications. However, it is hard to adopt the developed protocols directly translated to nanocellulosic substrates, especially on colloidal stable and highly charged cellulose nanofibers, such as TEMPO-oxidized and carboxymethylated CNF. In fact, we found only two works on NaIO_4 oxidation of charged CNF. In one of the works, the authors performed oxidation of carboxymethylated CNF in a dilute suspension and purified the product via dialysis lasting for 3 days (*Cellulose (2017) 24:3883–3899*). In another work, oxidation was performed on TEMPO-CNF hydrogel, isolated from the oxidation via centrifugation. It was used as a paper additive without characterizing the structural properties of oxidized nanofibers (*Carbohydrate Polymers 250 (2020) 116941*). To the best of our knowledge, the reported method is the first example of a simple, fast, and tunable oxidation of colloidal stable CNF. We managed to avoid cellulose aggregation and tedious separation by 1. using cellulose nanofibers containing hemicellulose (holo-CNF) 2. conducting the reaction on preformed holo-CNF wet cakes. In holo-CNF, cellulose nanofibers are covered by hemicellulose chains, which allows for efficient defibrillation with a small number of negative charges on the nanofibers' surface (e.g., TEMPO-CNF). The high content of negative charges during periodate oxidation slows down the oxidation kinetics, presumably via electrostatic repulsion of IO_4^- anions (which we refer to in the manuscript: "Using TEMPO-CNF in this transformation is impractical due to low reaction rate. The reason is repulsion of the IO_4^- anion by the negatively charged carboxyls at TEMPO-CNF surfaces." observed when trying to perform the reaction on TEMPO-CNF wet cake.

While hydroxylamine is used as a titrating agent to determine the number of aldehyde groups in dialdehyde celluloses, O-substituted hydroxylamines have not been implemented as functionalization reagents for cellulose materials. Generally utilized methods rely on the functionalization of oxidized cellulose with amines coupled with *in situ* reductions of the formed imines to secondary amines for hydrolytic stability. This work uses the increased hydrolytic stability of oximes vs. imines to perform direct functionalization without a reduction step while keeping the possibility for post-life defunctionalization. Moreover, using salts of O-substituted hydroxylamines allowed us to install even highly hydrophobic functionalities on oxidized CNF in water, which would be difficult to achieve otherwise. Other water-based systems for functionalization of CNF that are relevant to include silylation and amide coupling. A main disadvantage of amide coupling is the use of stoichiometric amounts of coupling agents; and the formation of a stable amide bond, making the transformation

irreversible. There are also several concerns regarding silylation, which can be performed in water. Trialkyl silanes are prone to hydrolysis and oligomerization, which result in less defined functionalization, where oligomeric siloxanes can be either covalently attached or absorbed on the surface of cellulosic material. In addition, due to several reactive sites on trialkylsilanes, it is challenging to control cross-linking (*ChemSusChem*, 2015, 8, 2681–2690, *Journal of Applied polymer science*, 2012, 125(4), 3084–3091). Moreover, the defunctionalization of the materials obtained by the above methods was not studied.

The presented method is extremely experimentally simple, where CNF wet-cake is directly placed in a water solution of a corresponding reagent, allowing for easy isolation of the modified material and direct reuse of the solution containing unreacted reagents.

We introduced substantial changes in the abstract and in the introduction of the manuscript to stress the novelty and advantageousness of the developed process.

2. Relying on NaIO_4 oxidation, the authors introduce a high amount of aldehyde groups on the CNFs ~4.5 mmol per g of CNF means 4.5 mmol aldehyde groups/6.2 mmol of cellulose are there, so 36% of AGU units have been oxidized. It is quite a high degree of oxidation. CNFs original structure is not preserved. I do not understand the usefulness of such an extensive skeleton functionalization. In the introduction, the authors declare that the challenges identified in this work include: “preserving the nanostructure of CNF, enabling high accessibility of the functional groups, and overcoming moisture sensitivity of the transformation”. Indeed, it is accepted that a low degree of substitution/functionalization ($DS < 0.37$) will preserve the nanomorphology of nanocellulose (D.-Y. Kim, Y. Nishiyama, and S. Kuga, “Surface acetylation of bacterial cellulose,” *Cellulose*, vol. 9, no. 3–4, pp. 361–367, 2002). I think the authors should discuss more and perform the necessary characterizations also on samples with lower oxidation extent, but more preserved cristallinity.

Response: We thank the reviewer for raising this crucial point. Indeed, 4.5 mmol/g is a very high degree of oxidation, where the oxidation proceeds not only on the surface of CNF. We performed the reaction on a highly oxidized substrate for our substrate scope (salts of O-substituted hydroxylamines) mainly to make the quantification of the transformation more accurate. However, as shown in Figure 1 and Table S1, the extent of the oxidation is easily tuned by changing the oxidant concentration or the reaction time. We examined the properties of the film with a lower degree of oxidation (in the manuscript, we refer to it as DA-CNF-2h). In addition, we performed the functionalization of this film (Bn-ON-CNF-2h) and defunctionalization via hydrolysis. We present the obtained results in Figure 3c.

To clarify, we mention the reason and choice of such a high degree of substitution in the main text of the manuscript (highlighted in green):

“In addition, for substrate scope studies, the reaction was driven to high DF to facilitate characterization of the final products. However, the designed methodology allows for simple modulation of DF, where lower DF (mainly surface modification of nanofibrils) is desirable when mechanical properties of the material are of the greatest importance.”

3. Microscopy investigation of the functionalized samples needs to be performed, either by AFM or scanning electron microscopy.

Response: We have now added SEM and SEM/EDS images of cross-section of Bn-ON-CNF film to show the structure and to demonstrate that modification occurs homogeneously within the thickness of the film (Figures S8 and S9).

Figure S8. SEM image of Bn-ON-CNF film (cross section).

Figure S9. SEM/EDS images of Bn-ON-CNF films (cross section) showing distribution of carbon (left) and nitrogen (right).

4. Reversibility: how is the appearance of fibers after the removal of oximes?

Response: The defunctionalized films appeared colorless when the reaction was performed in an acetone/water mixture at room temperature. Slight coloration occurred when the film was further treated under heating for the complete removal of the functional group. We add the following statement to the main text of the manuscript (highlighted in green):

“The defunctionalized films appeared colorless when the reaction was performed in acetone/water mixture at room temperature. Slight coloration occurred when the film was further treated under heating for the complete removal of the functional group.”

5. How do the authors know that CNFs are composed of 18 cellulose chains? Are there experimental data to support this statement?

Response: The exact built-up of native wood and CNF is an ongoing debate, especially the number of cellulosic chains in the elemental fibril. As this study doesn't address these questions, we removed this statement from the manuscript.

“Each nanofiber is semicrystalline and consists of more than 18 parallel cellulose molecules in extended chain conformation.” Was modified to “. Each nanofiber is semicrystalline and consists of parallel cellulose molecules in extended chain conformation”

6. The first scheme of supporting information is wrong: the correct product is hydroxylamine hydrochloride, not amine hydrochloride.

Response: We thank the reviewer for mentioning this mistake. We have now introduced corresponding changes in the Supporting information.

7. Please explain the following sentence: "In the case of more rigid hydrophobic substituents (e.g., R1-R5, R10), the R-ON-CNF wet cake stiffens during the progression of the reaction. In contrast, Me- and pentyl-functionalized DA-CNF wet cakes stayed soft and flexible."

Response: We agree that the statement needs to be rephrased. We change it to "In the case of substituents R1-R5, R8-R10, the R-ON-CNF wet cake stiffens during the progression of the reaction. In contrast, Me- and pentyl-functionalized DA-CNF wet cakes stayed softer and more flexible". We highlight the corresponding changes in green in the manuscript.

8. "Whether the difunctional hydroxylamine reacted with one or two aldehydes was assessed by using triflates as a counterion instead of halides." It would be much more reliable to look at the bifunctional reagents by XPS technique.

Response: We agree that additional studies need to be performed to allow us to make this statement. As such, this statement has been removed in the new version of the manuscript.

9. The authors decided to study kinetics of oxime removal following CF₃-phenyl group dissolution by NMR spectroscopy. They also report an example with a pentafluorobenzene appendage on oximes. In my experience these rings are susceptible to nucleophile attack, did the authors check their water stability?

Response: We have now thoroughly evaluated the stability of the standard used in our study (pentafluoro benzyl bromide) under the reaction conditions. We did not observe any degradation of the standard. We include NMR spectra illustrating the stability the standard under applied conditions (Supporting information, page 35).

Evaluation of the stability of the standard (pentafluoro benzyl bromide) under reaction conditions. Top: ^{19}F NMR spectrum of pentafluoro benzyl bromide treated in 1M(HCl) acetone/water mixture for 96 hours at room temperature. Bottom: ^{19}F NMR spectrum pentafluoro benzyl bromide.

Reviewer #2

This is an interesting paper, showing the preparation and characterization of DA-CNF and various R-ON-CNFs, and application of R-ON-CNFs to triboelectric generators, from some new aspects. Therefore, this manuscript can be publication in Nature Communications after revisions.

1) Since the R-ON-CNFs still contained some aldehyde groups, not only DA-CNF films but also R-ON-CNF films may be discolored after drying at 93C. This discoloration of aldehyde-containing films should be addressed in terms of aldehyde content.

Response: We have not observed any coloration of the prepared films, original holo-CNF, oxidized (DA-CNF), or functionalized R-ON-CNF. For clarification, the following Figure S7 has been added to supporting information.

Figure S7. Pristine and modified CNF films on the surface (left) and lifted a few cm from the surface (right) for a. Holo-CNF, b. DA-CNF, c. Bn-ON-CNF.

2) It may be difficult in this study, but the yields (based on the starting dry CNF weight) and molecular weights of DA-CNFs prepared under various conditions should be measured or discussed. DA-CNFs may partly decrease in the yield by oxidation and washing processes.

Response: We agree with the reviewer that it would be exciting to see the changes in molecular weight as this would give us intel into the mechanistic features of the oxidation, possibly answer the question on is the hemicellulose oxidized before cellulose, etc. We tried to evaluate this with SEC, however, we had real issues obtaining reliable data due to issues dissolving the films according to standard protocols. The only data point we could get was after 4.5 h of oxidation, see figure below.

This data suggests that during oxidation, we have degradation of the cellulose substrate. However, this is still too uncertain to draw this conclusion. For one, oxidation leads to the hemiacetal formation, which undoubtedly will lead to intra-molecular acetal-formation that changes the hydrodynamic volume. In addition, we had problems with solubilizing the sample. As such, we are unsure which fraction we analyze, i.e., is the high molecular fraction not dissolved? To answer these questions, we need to develop a new solubilizing system and combine that with both NMR and viscosity measurements, which we believe falls outside of this study's scope.

Regarding to the yield of the oxidation step, we refrain of such statements in the manuscript, because of necessity to perform the reaction at higher scales to provide reliable and reproducible data. Our preliminary results revealed that 85-95 wt% of the material can be recovered.

3) Not only the conversion ratio from aldehyde groups to oximes (i.e., 73% at 1.5 eq. of Bn-ONH₃+Cl- in page 5) but also the efficiencies of oxime formation (i.e., oximed amine/added amine by mol) should be added in Table 1).

Response: We have now added the requested information in Table 1 (highlighted in green in the manuscript).

4) It is better also to describe the reaction efficiency of NaIO₄ in the DA-CNF preparation. Namely, the molar ratio of NaIO₄ consumed for aldehyde formation/that added in the oxidation.

Response: We agree with the reviewer that consumption of NaIO₄ can also be used to determine the degree of oxidation of dialdehyde cellulose. However, it has been shown that hydroxylamine titration and NaIO₄ consumption methods show a relatively good correlation. In light of this, we limited ourselves to only the hydroxylamine titration method (*Thermochim. Acta* 2001, 369 (1–2), 79– 85).

5) Page 7: The reaction conditions in acidic acetone/water at 55 degree C for 24 h may be harsh for R-ON-CNF. The low yield losses (<1%) were described in page 7, but some depolymerization of oxidized cellulose molecules may be unavoidable during de-oxime reactions, which should be addressed.

Response: Indeed, partial depolymerization of cellulose may occur under acidic conditions. However, this factor is hard to assess due to problems with determining the changes in molecular weight by SEC. However, we did not observe any film coloration after oxime hydrolysis reactions, which is an indirect evidence of a low degree of cellulose degradation. We have added the following statement to the manuscript (highlighted in green):

“Even though partial depolymerization of cellulose may occur under aqueous acidic conditions, we did not observe any coloration of the film after oxime detachment, indicative of limited cellulose degradation.”

6) Figure 1: Since C.I decreased to almost zero after oxidation for 7.5 h, did this oxidation occur also in the crystalline inside CNFs without reaction selectively on the CNF surfaces?

Response: Yes, we interpret this data as oxidation occurs inside CNF and not only on the surface at prolonged reaction times. We have added the following statement in the manuscript (highlighted in green):

“We found that after 7.5 hours, crystallinity of the samples was completely lost, suggesting that the oxidation occurred not only on the surface of CNF, but within inner regions as well.”

7) Figure 2, green spectrum: Is it true that the C-H peak is located at ~3250 cm⁻¹?

Response: Yes, according to the literature (Socrates G. *Infrared and Raman characteristic group frequencies: tables and charts, 3rd ed. edn. J. Wiley (2001)*) C-H stretching in alkynes occurs in the region of 3200-3400 cm⁻¹.

8) Figure 3b: The changes in water uptakes were consistent with water contact angles on the films?

Response: We agree with the reviewer that it is a crucial point to mention. We chose to assess water uptake rather than contact angles because hydrophobicity is a bulk property and is better reflected by water uptake measurements than contact angle measurements. This relates to the fact that the surface roughness of the samples has a significant effect on contact angles, making conclusions regarding the samples' hydrophobicity difficult to draw. We add the following statement to the manuscript (highlighted in green):

“Water sorption of cellulosic materials is a bulk property; therefore, we studied the water uptake for CNF, DA-CNF, and some R-ON-CNF films rather than water contact angles (which can also be influenced by the surface roughness), presented in the table on Figure 3b”.

9) Figure 3a, S4, S5, and S6: It is better to include photographs of the films in this manuscript to show the colors or transparencies.

Response: We have now included requested pictures to Supporting information (Figure S7).

Figure S7. Pristine and modified CNF films on the surface (left) and lifted a few cm from the surface (right) for a. Holo-CNF, b. DA-CNF, c. Bn-ON-CNF.

10) Figure S6: Why the films have such high haze values? Many pores with air/material interfaces are present in the films? What are the factors influencing the haze values of the films?

Response: The haze can be caused by surface roughness and by the presence of pores in the material. As can be seen from Figure S6, by preparing samples with smoother surfaces, haze can be significantly decreased (e.g., Me-ON-CNF haze reduces from 80 to 11% for the samples with rough and smooth surfaces, respectively, Table S3).

11) Figure S1: What is the signal at ~50 ppm owing to?

Response: Signal around 50 ppm is a spinning side band originating from the signal of carbon atoms in the aromatic ring. The main band of these protons appears in the region 125-130 ppm.

12) Figure S1 and Figure 2c: Since the signal pattern at 60-110 ppm in Figure S1 is similar to that in

Figure 2C, this sample may have significant amounts of aldehyde groups (or the oximation ratio may be too low). Please explain this matter. Or it is better to show another C13-NMR spectrum of R-ON-CNF with a much higher degree of oximation.

Response: We apologize for the confusion. Figure S1 and the right-hand side of Figure 2c are identical spectra, but Figure S1 also shows signal assignment. Figure 2c shows two NMR spectra. The left spectrum corresponds to DA-CNF, and the spectrum on the right side of the figure corresponds to Bn-ON-CNF. There is a striking difference between the two spectra (appearing of new signals corresponding to the aromatic ring and oxime linkage and a decrease of intensity of the region corresponding to oxidized cellulose 60-110 ppm).

13) Signal assignments should be added to Figure S3 and all 1H- and 13C-NMR spectra in Appendix. The solvents used in the 1H- and 13C-NMR measurement are also better to be described in the figures.

Response: We thank the reviewer for this comment. We agree that signal assignment would be very beneficial, however, in this case, a scaling up of the process would be necessary to enable the isolation of all the products. Since only a crude mixture was assessed for the reaction presented in Figure S3, we would like to refrain from the full assignment and will highlight the presence of the product resulting from the trapping of the released O-substituted hydroxylamine with acetone (signals around 1.9 ppm). We also include the information regarding the solvent used.

Reviewer #3

The present manuscript with the title “In water synthesis of highly functional, hydrophobic, and chemically recyclable cellulose nanomaterials through oxime ligation” describes an interesting study on the chemical modification of nanostructured cellulosic materials. The manuscript is well structured, the work concisely described, the experimental procedures can be repeated based on the protocols and the illustrations are of satisfying quality.

Nevertheless, I believe that the work is not suited for publication in Nature Comm. There are two main reasons. First, the element of novelty, a prerequisite to publication, is largely missing. As the authors state themselves, periodate oxidation of carbohydrate (the very old Malaprade oxidation) is very well-known and has been extensively applied also to (nano)cellulosic materials. In most cases, those studies did not target the “dialdehyde cellulose” directly, but used it as an intermediate for further functionalization. Schiff bases as well as non-substituted and substituted hydrazones and oximes have been synthesized from dialdehyde cellulose and other dialdehyde polysaccharides. Despite placing the emphasis more on the properties of the products, the manuscript does not offer any novel aspects here.

Response: We appreciate the possibility of clarifying some of the concerns expressed by the reviewer regarding the novelty and advantages of the developed method. Although NaIO₄ oxidation of cellulose fibers is a known process, NaIO₄ oxidation of nanocelluloses is far less studied. Oxidation directly on nanocellulose fibers increases the degree of oxidation (due to a large amount of OH-group exposed) and the homogeneity of aldehyde groups within the substrate. Unlike cellulose fibers, where inner parts of fibers are less available for oxidation, cellulose nanofibrils are individualized, allowing for better chemical accessibility. While there are reports regarding the oxidation of cellulose nanocrystals (CNC), the oxidation of cellulose nanofibers (CNF) is very rare. We found only two papers on NaIO₄ oxidation of CNF; in both cases, CNF was pre-modified either via carboxymethylation or TEMPO oxidation. The oxidized materials were used as additives to improve the properties of papers rather

than studied as individual materials. In addition, separation of the product from the solution of an oxidant is difficult (dialysis for several days was implemented in one of the works). In our work, we developed a novel and direct method to perform fast, efficient, and experimentally simple oxidation of CNF. The key parameters for the successful oxidation are: 1) use of CNF with low charge (holo-CNF vs. TEMPO CNF) to avoid anionic repulsion between IO_4^- and COO^- groups on cellulose, which we observed when trying to perform the oxidation on TEMPO-CNF wet cake; 2) perform the reaction on the preformed wet cake to avoid separation problems and aggregation of CNF. To our knowledge, O-substituted hydroxylamines have not been implemented as functionalization reagents for cellulose modification. Most reported methods used amines to produce imines followed by (or performed in situ) reduction. In this work, we raised a scientific question of whether improved hydrolytic stability of oximes will remove the stabilization step via reduction while still providing stable covalent attachment. Our results show successful covalent attachment of various functional groups to CNF. Moreover, a complete detachment of the installed functional groups was performed under mild reaction conditions.

It is essential to mention that both steps are very experimentally simple. The CNF wet cake is placed into a beaker containing water solution of either oxidant (first step) or a corresponding salt of O-substituted hydroxyl amine (second step), allowing for simple product isolation and removal of the unreacted derivatizing agent.

Second, there are several aspects of the work which seem to indicated the studies have not yet fully matured (see below), and there are several claims which seem a bit exaggerated.

- line 3 (title): the products are certainly less hydrophilic than cellulose, but to call them hydrophobic would be quite an exaggeration (cf. for instance to hydrocarbons). The product still possesses a wealth of hydrophilic functions, and in case of the hydrochlorides or other salts are even ionic.

Response: Yes, in the final material, there are still hydrophilic functional groups, which come from the unreacted cellulose fragments. However, our studies on water uptake demonstrated that some substrates were significantly hydrophobized (water uptake decreased from 231% for CNF to 6% for pentafluoro functionalized CNF, Figure 3b). With regards to hydrochlorides, these are not present in the final materials since salts of O-substituted hydroxylamine hydrochlorides form uncharged oximes once reacted with aldehyde groups of dialdehyde CNF.

- line 76: not all modifications of CNF suffer from moisture sensitivity. There have been general approaches to reactions of CNF proceeding in the surface-confined water layer (Beaumont et al. Nature Comm. 2021) or at least being fully water-tolerant (Hettegger et al. Chem.Sus.Chem. 2015 and 2016).

Response: We appreciate that the reviewer brings up this point. True, some methodologies suggest it occurs in a water medium. For example, in the presented paper by Beaumont et al. *Nature Comm. 2021* et al. on acetylation in the surface-confined water layer. However, this method cannot be applicable to the modification of CNF wet cakes (ca. 85 wt%). In addition, stating that silane chemistry is fully water-tolerant is inaccurate. Silanes are known to react in the presence of water, forming Si-OH substrates that undergo further condensation, resulting in the formation of nano-clusters. (*Colloids and Surfaces A: Physicochem. Eng. Aspects 312 (2008) 83–91*) Comparably, oxime ligation is high yielding regardless of water content. The chemistry is orthogonal, meaning it is the intended reaction that occurs, and no other side reaction that might be misinterpreted as a successful reaction outcome.

The following alterations to the manuscript have been made to clarify this and adhere to the reviewer's comments.

"...For these transformations solvent exchange is generally required, even though there are acetylation methods which tolerate residual water."

"Silylation, while being promising example of water-based modification pathway, can be complicated due to the propensity of trialkyl silanes to hydrolysis and oligomerization which can result in less defined functionalization, where oligomeric siloxanes can be either covalently attached or absorbed on the surface of cellulosic material."

- line 89: it is not fully clear what "prefunctionalization" means. TEMPO oxidation, periodate oxidation or acyl transfer catalyzed by imidazole would not require any preceding steps.

Response: We apologize for the confusion caused by this statement. We have now removed it from the manuscript.

- line 89: the statement regarding atom economy would require a solid support by numbers to demonstrate that the proposed periodate oxidation / oxime approach is superior to the alternatives.

Response: We refer to atom economy regarding amide coupling, which is another method to perform modifications of oxidized CNF (TEMPO-CNF) in water. These protocols require stoichiometric amounts of coupling agents (e.g., 1-Ethyl-3-(3-dimethylaminopropyl)carbodiimide, EDC), which produces a stoichiometric amount of waste. While in the case of oxime formation, the only by-product is HCl. We have introduced the following clarification in the manuscript (highlighted in green):

"However, these methods require stoichiometrically matched amounts of reagents (EDC, NHS) and generate the stoichiometric amount of wastes, which complicates the purification process and compromises atom efficiency of the transformation."

- line 91: the statement of "harsh conditions" needed for functional group / substituent removal is not fully conclusive: a 0.1 M NaOH used for removal of acetyl/acyl groups is not harsher than the 1 M HCl used in the present work to cleave the oximes.

Response: We agree that esters can be cleaved under relatively mild reaction conditions. We mainly refer to the cleavage of amide bonds (which can be formed in water via amide coupling). We have now modified the statement in the main text of the manuscript (highlighted in green):

"Thus, harsh reaction conditions are needed to detach functional groups from surface-modified CNF (e.g., hydrolysis of amides) or a stoichiometric reagent (e.g., fluoride source for silyl ethers). The possibility of mild "on-demand" detachment of the covalently linked functional groups would enable recycling of the chemicals and significantly improve the sustainability and economic feasibility of modified CNF in targeted applications."

- line 105: this statement is not supported by a literature survey: the "dialdehyde" structure obtained by periodate oxidation is in very many cases used to have a reactive moiety for further modification (mostly by reacting with N- or C-nucleophiles). This further modification of the aldehydes is certainly not "less common".

Response: We have now removed this statement from the manuscript.

- line 112: current reduction methods rather use aminoboranes which are selective towards imines and leave the aldehydes/hemiacetals unaffected.

Response: We fail to find the corresponding literature on using aminoboranes as a selective reductant of imines over aldehydes in water. However, we found literature that shows that aminoboranes are very good reductants for aldehydes and ketones in water (*Selective reduction of aldehydes and ketones to alcohols with ammonia borane in neat water*[†] Lei Shi, Yingying Liu, Qingfeng Liu, Bin Wei and Guisheng Zhang* DOI: <https://doi.org/10.1039/C2GC00006G>) Regardless, due to the low hydrolytic stability of imines, the reported procedures for functionalizing aldehyde groups on dialdehyde celluloses with amines to imines necessitate reduction. Although, it might be possible to maximize the formation of the desired amines and minimize the formation of alcohols from aldehyde reduction by careful modulation of pH, excess of amine, and even the use of specific catalysts. In this protocol, oxime ligation does not require any additional steps. It leads to the formation of stable C-N bonds, which can be cleaved on-demand in the presence of a trapping agent under acidic conditions.

- line 114: the evolution of hydrogen gas is strongly depending on the conditions and can be largely avoided by deeper temperatures, higher pH, alcoholic co-solvents, etc.

Response: We agree that side reactions of NaBH₄ and related reducing agents with protic species can be minimized via optimization of the reaction conditions. However, reduction is still an additional step and still leads to the formation of stable products difficult to defunctionalize. We introduce the following changes in the manuscript (highlighted in green):

“Reduction step is necessary due to the hydrolytic instability of imines in water. In addition to apparent disadvantages of an extra step, in situ reduction of imines is generally complicated by the competing reduction of aldehydes, resulting in a low degree of functionalization (e.g., 11 – 24% of aldehyde groups were modified). The reaction is exothermic, with hydrogen gas as a possible byproduct, even though via an optimization of the reaction conditions this side reaction can be minimized. And importantly, the generated amine group is highly stable, limiting both the recovery and recycling of the functionalized CNF”.

- line 122: it is correct that for oxime cleavage often ketone or aldehyde traps are used to scavenge the release hydroxylamine. However, these traps are bound to an appreciable extent by the dialdehyde polysaccharides. If this occurs by hemiacetal bonds, the reaction is reversible and the binding temporary, if covalent bonds are formed (acid-catalyzed aldol), the scavenger is consumed and the (surface) structure and chemistry of the polysaccharide irreversibly altered. This chemistry thus needs much further work to become fully reliable.

Response: We agree with the reviewer that using an aldehyde as a trapping agent could lead to forming acetals with cellulose through hemiacetal formation. As such, we instead use acetone as a trap in our protocol. Instead of aldehydes, acetone would form ketals with cellulose. ketals are much more unstable than acetals (*J. Am. Chem. Soc.* 2017, 139, 6, 2306–2317), and we do not see any evidence of this formation.

We do not see evidence of aldol condensation between aldehyde groups and acetone under our conditions. Aldol condensation would form α,β unsaturated bonds, evident by the ¹³C NMR spectrum of the defunctionalized Bn-ON-CNF film (SI, page 33). We have not observed any of these signals, as well as ketones or methyl groups, which would indicate the covalent attachment of acetone to CNF.

Defunctionalized Bn-ON-CNF-2h

- line 143: the amount of acetal linkages is rather low. Moreover, if they exist they would not react in the proposed way with hydroxylamines.

Response: In literature studies of dialdehyde cellulose, it has been concluded that aldehydes are mainly involved in forming hemiacetal/acetal linkages. For example, Leguy et al. claim (ACS Sustainable Chem. Eng. 2019, 7, 412–420): “As systematically reported, the absence of signals in the 160–200 ppm region indicates that the aldehyde groups do not exist in their free form but have readily recombined with neighboring hydroxyl groups to give various hemiacetal and/or hemiacetal entities, which give rise to a broad signal in the 90–100 ppm region.” and support their statement by several references (Compos. Sci. Technol. 2015, 117, 54– 61; Biomacromolecules 2014, 15 (5), 1928– 1932; Cellulose 2014, 21 (6), 4119– 4133; Biomacromolecules 2016, 17 (9), 2972– 2980; Cellulose 2015, 22 (3), 1743– 1752; Cellulose 2017, 24 (7), 2753– 2766).

Under acidic aqueous conditions, there is an equilibrium between aldehyde and acetal/hemiacetal forms of dialdehyde cellulose, allowing for oxime ligation.

We added the following statement to the manuscript (highlighted in green):

“An absence of the signal corresponding to C=O group of dialdehyde cellulose due the formation of hemiacetal/acetal linkages was well documented in previous studies”

- line 157: the use and fate of the “CNF wet-cake” is not fully clear. Why would it not disintegrate in the aqueous surroundings? Is the contact time with the periodate reagent (and the resulting extent of oxidation) everywhere in the cake the same?

Response: CNF wet cake does not disintegrate upon the treatment and can be easily removed from the solution and washed on filter paper. In addition, as we discuss in the manuscript the wet cake shrinks in xy direction and thickens in z direction, making it even easier to handle after the oxidation. After oxime ligation, we performed SEM/EDS mapping analysis of the derivatized films to assess how evenly N atoms (deriving from oxime linkages) are distributed within the film (Figure S9). We observed an even distribution of the functional groups within the sample, indicating that both steps (oxidation and oximation) occur homogeneously.

- line 183: the problem with increasing iodate concentrations is also the increasing instability of periodate (of which the mechanism is unknown)

Response: As we demonstrated in Table S1, the reaction can be performed under higher and lower concentrations of NaIO₄, and tuned by the reaction time. We did not observe any issues when the reaction was performed with a high concentration of NaIO₄.

- lines 209-212: the reason for the selection of these substituents remains somewhat unclear. It appears as if just the commercially available O-substituted hydroxylamines were tested. This demonstrates the general applicability of the procedure, but would not support claims that there was any “targeted” selection.

Response: We apologize if this was not clear from the manuscript's text, but only two out of ten tested O-substituted hydroxylamines were commercially available (Bn-ONH₃⁺Cl⁻ and Me-ONH₃⁺Cl⁻). Others were synthesized, and the synthetic procedures are reported in Supporting information, pages 2-4. To clarify, we include the following statement in the manuscript (highlighted in green):

“The corresponding salts of O-substituted hydroxylamines (apart of commercially available Bn-ONH₃⁺Cl⁻ and Me-ONH₃⁺Cl⁻) were synthesized from hydroxylamine and corresponding bromides, while synthesis from alcohols is also possible.”

- line 220/387 (e.g.): the reagents should be called O-substituted hydroxylamines, but not “ether of hydroxyl amine”

Response: In order to distinguish reagents used in our work (ethers, R-ONH₃⁺X⁻) from esters (R-C(O)-ONH₃⁺X⁻) of hydroxylamines, we sometimes refer to them as ethers.

- lines 270/276/281-282: the “recycling of CNF” is quite an exaggeration. First, after cleavage, the product is not cellulose anymore, but periodate oxidized cellulose – the second modification step, the oxime formation, might be reversible, the first one, periodate oxidation, is not. Second, the conditions required for oxime cleavage (1M HCl, hydroxylamine scavengers), are rather detrimental to cellulose integrity. The acidity causes hydrolytic cleavage, which is already well noticeable at that pH and reaction times, and the scavenger might react with the dialdehyde cellulose and modify it (see above).

Response: We agree that the recovered CNF is not pristine from the perspective of the initial holo-CNF but rather as before oxime ligation. Satisfyingly, ¹³C NMR analysis (Supporting information, page 33) and FTIR experiment (Scheme 3b) showed that the initial structure of dialdehyde CNF was preserved to a large extent. In addition, the mechanical properties of the defunctionalized films further support that the structural integrity of the films can be maintained (Figure 3c). The defunctionalization simplifies the post-life treatment of CNF films, where functional substituents and dialdehyde CNF films can be recovered and reused.

To clarify the following statement has been added to the manuscript (highlighted in green): "It is important to mention that upon the defunctionalization DA-CNF, rather than the pristine CNF is recovered."

- line 396: is such a high degree of oxidation / modification really compatible with the claim that only the surface of the CNF is modified?

Response: Surely, a high degree of oxidation (4.8 mmol/g) of CNF means that functionalization occurred not only on the surface of nanofibrils but throughout the entire material. Also clearly seen in the reduction of cellulose crystallinity at a high degree of oxidation. As we stated in the manuscript, a high degree of oxidation was used to simplify the quantification of the transformation. However, our method allows for simple tuning of the degree of oxidation via either concentration of the oxidant or reaction time (Table S1).

We introduced the following statement in the manuscript (highlighted in green):

"In addition, for substrate scope studies the reaction was driven to high DF to facilitate characterization of the final products. However, the designed methodology allows for simple modulation of DF, where lower DF (mainly surface modification of nanofibrils) is desirable when mechanical properties of the material are of the greatest importance."

- line 410: the experiments used holocellulose, which contains certain amounts of hemicelluloses which are more reactive than cellulose in periodate oxidations. This aspect is not considered, but it might imply that a certain amount of modified CNF is actually modified hemicellulose.

Response: We thank the reviewer for the insightful comment. Attempts have been made to assess which portion of hemicellulose and cellulose oxidized as the hemicelluloses are also prone to oxidation. We attempted to assess this through SEC. Unfortunately, there were problems with solubility, making it hard to conclude what was analyzed (i.e., is the high molecular fraction not dissolved). To answer this with high accuracy, we need to develop a new solubilizing system combined with NMR and viscosity measurements that fall outside this study's scope.

To clarify, the following statement has been added to the manuscript (highlighted in green):

"It is important to mention that part of aldehyde groups can be located on hemicellulose chains, which cover cellulose nanofibrils in holo-CNF."

REVIEWERS' COMMENTS

Reviewer #1 (Remarks to the Author):

I feel satisfied with the explanations offered by the authors, justifying the motivation of the work. The paper now can be considered for publication in this journal. As a suggestion for further improving the manuscript: SEM investigation can be supported by micrographs with higher magnification.

Reviewer #3 (Remarks to the Author):

The authors have answered most of the questions in a thoughtful manner. While some details have been solved, I see the main objection, the lack of novelty of the work, persisting. Both the malaprade oxidation itself and the modification of the aldehyde intermediates are well known and practiced very frequently in carbohydrate and also (nano)cellulose science. A simple variation of the substrates does not provide enough novelty to warrant publication in Nature Communications. The submitted work has certainly been done properly and provides a range of cellulose derivatives (with different substituents), but from the point of view of novelty and generality it should be published as an original article in a journal of materials science or carbohydrate chemistry, but not as an urgent, novel account.

I also consider some other more technical issues to be unresolved. The claim of "recycling" may be true for the dialdehyde cellulose stage (after splitting off the oximes), but certainly not for cellulose itself. Periodate cellulose, regardless of the degree of oxidation, is substantially different chemically from the underlying cellulose. The question of the "wet cake" remains open. Why does this cake not simply disintegrate upon reaction and washing? The question of the role and reactivity of the hemicelluloses also remains unanswered. These "uncertainties" in the paper are excusable for papers in standard journals, but would not be acceptable for a publication in Nature Commun.

Answers to the Reviewers

Reviewer #1 (Remarks to the Author):

I feel satisfied with the explanations offered by the authors, justifying the motivation of the work. The paper now can be considered for publication in this journal. As a suggestion for further improving the manuscript: SEM investigation can be supported by micrographs with higher magnification.

Answer: We thank the reviewer for the positive response. Indeed, SEM investigation with micrographs at higher magnification will be carried out in future studies.

Reviewer #3 (Remarks to the Author):

The authors have answered most of the questions in a thoughtful manner. While some details have been solved, I see the main objection, the lack of novelty of the work, persisting. Both the malaprade oxidation itself and the modification of the aldehyde intermediates are well known and practiced very frequently in carbohydrate and also (nano)cellulose science. A simple variation of the substrates does not provide enough novelty to warrant publication in Nature Communications. The submitted work has certainly been done properly and provides a range of cellulose derivatives (with different substituents), but from the point of view of novelty and generality it should be published as an original article in a journal of materials science or carbohydrate chemistry, but not as an urgent, novel account.

I also consider some other more technical issues to be unresolved. The claim of "recycling" may be true for the dialdehyde cellulose stage (after splitting off the oximes), but certainly not for cellulose itself. Periodate cellulose, regardless of the degree of oxidation, is substantially different chemically from the underlying cellulose.

Answer. We agree with the comment on recycling, and the following statements have been added to the new version of the manuscript to clarify this.

Abstract. Page 1 - Before

"The process provides a high degree of nanofibril functionalization (2 – 4.5 mmol/g) and allows for the reversible detachment of the functionality under mildly acidic conditions."

New version

"The process provides a high degree of nanofibril functionalization (2 – 4.5 mmol/g) and allows for the reversible detachment of the functionality under mildly acidic conditions, resulting in the reformation of dialdehyde CNF."

Results and Discussion – Page 7 - before

"The possibility to selectively remove functional groups on cellulose nanofibers is of great importance, allowing for the recovering of the functional moiety and recycling of CNF."

New version

“The possibility to selectively remove functional groups on cellulose nanofibers is of great importance, allowing for the recovering of the functional moiety and recycling of CNF. It is important to mention that upon the defunctionalization DA-CNF is recovered, rather than the initial holo-CNF. However, several studies demonstrated that DA-CNF is also biodegradable, meaning that the post-life treatment of defunctionalized DA-CNF is significantly easier compared to the functionalized CNF material.”

Conclusions – Page 10 - before

The complete detachment was achieved with acetone as a trapping agent at ambient temperature, allowing for recovery of both CNF and the corresponding functional moiety.

New version

“The complete detachment was achieved with acetone as a trapping agent at ambient temperature, allowing for recovery of both dialdehyde-CNF and the corresponding functional moiety.”

The question of the "wet cake" remains open. Why does this cake not simply disintegrate upon reaction and washing? The question of the role and reactivity of the hemicelluloses also remains unanswered. These "uncertainties" in the paper are excusable for papers in standard journals, but would not be acceptable for a publication in Nature Commun

Answer. Instead of disintegration of the “wet cake” we rather see that it shrinks in the xy direction and becomes thicker in the z direction in accordance with the degree of oxidation, see figure below

The shrinkage of the wet cake indicates that some chemical interaction occurs within the sample that gives rise to this, possibly causing cellulose nanofibril contraction in x-y plane. The chemical interaction was further verified by solid-state carbon NMR. After oxidation, this gives rise to the formation of aldehyde. Aldehydes have a chemical shift of around 200 ppm; as seen in the figure below, none of these

peaks are present in the material, indicating that most of the aldehydes are either hydrated or participate in hemiacetal formation with the remaining hydroxyls, see figure below.

This observation follows what has been seen in the literature (references are present in the manuscript). As such, upon oxidation, a reversible cross-linked CNF network is formed, which is believed to be one of the primary reasons we do not observe any disintegration of the "wet cake."

To clarify the following statement has been added to the new version of the manuscript.

Added to the new version of the manuscript

"Even at a high degree of oxidation, none of the aldehyde groups were observed by solid-state carbon NMR. The absence of shifts above 200 ppm suggests that the aldehydes are hydrated or participate in hemiacetal formation with residual hydroxyls in the carbohydrates. As such, shrinkage is believed to be due to a combination of hemiacetal formation and transition from strong in-plane CNF orientation to a state where CNF fibrils contract in-plane and possibly become wavy with out-of-plane oriented segments (thickness increase)."

Regarding the role of the hemicelluloses.

Besides that, remaining hemicelluloses facilitate the "wet cake" formation according to our previous work, <https://pubs.acs.org/doi/full/10.1021/acsnano.9b07659>. There might also influence the oxidation kinetics. As observed when compared to TEMPO-CNF, holo-CNF gives a much faster reaction. One part is the charge repulsion of TEMPO-CNF that makes the oxidation kinetics slower, whereas the other part is the chemical accessibility of the hemicellulose in the system. We tried to evaluate in which order the reaction proceeds by measuring the changes in molecular weight, as hemicelluloses and cellulose have

different distributions. Due to problems with the solubility of the samples (due to hemiacetal formation), we could not draw any solid conclusions. Indirectly we can observe that the reduction in crystallinity of cellulose follows two different regimes; see figure below.

Suppose the reactivity between cellulose and hemicellulose are the same, leading to a linear dependence between the conversion and decrease in crystallinity, which is not observed here. To clarify this, the following statement has been added to the new version of the manuscript.

"In summary, it is believed that the hemicellulose-covered surface of holo-CNF both facilitates chemical accessibility and rate of oxidation. The reduction rate in crystallinity of cellulose follows two different regimes with oxidation conversion, where the rate is lower at the beginning, followed by sharp increase after 25% carbohydrate conversion. A possible interpretation is that the hemicelluloses are primarily oxidized first then, followed by cellulose oxidation."